# Development, Characterization, and Clinical Investigation of a New Topical Emulsion System Containing a *Castanea sativa* Spiny Burs Active Extract

**DOI:** 10.3390/pharmaceutics13101634

**Published:** 2021-10-07

**Authors:** Tiziana Esposito, Teresa Mencherini, Francesca Sansone, Giulia Auriemma, Patrizia Gazzerro, Rosa Valentina Puca, Raffaele Iandoli, Rita Patrizia Aquino

**Affiliations:** 1Department of Pharmacy, University of Salerno, 84084 Fisciano, Italy; tesposito@unisa.it (T.E.); gauriemma@unisa.it (G.A.); pgazzerro@unisa.it (P.G.); aquinorp@unisa.it (R.P.A.); 2Unesco Chair Salerno, Plantae Medicinales Mediterraneae, University of Salerno, 84084 Fisciano, Italy; 3COSM-HI Lab,“San Giuseppe Moscati” National Hospital (AORN), Contrada Amoretta, 83100 Avellino, Italy; 4Dermatology and Dermosurgery, “San Giuseppe Moscati” National Hospital (AORN), Contrada Amoretta, 83100 Avellino, Italy; dott.puca@gmail.com (R.V.P.); Raffaele.iandoli@gmail.com (R.I.)

**Keywords:** chestnut spiny burs extract, O/W emulsion system, clinical trials, skin compatibility, in vivo skin performance, self-assessment test

## Abstract

The study focused on the development and characterization of an O/W emulsion for skincare containing *Castanea sativa* spiny burs extract (CSE) as functional agent. The emulsion was stable and had suitable physicochemical and technological properties for dermal application and CSE showed no cytotoxicity in spontaneously immortalized keratinocytes (HaCaT) at active concentrations. A single-blind, placebo-controlled, monocentric study was designed to evaluate the skin tolerability and the skin performance of the CSE-loaded emulsion on healthy human volunteers. An improvement was observed in skin biomechanical properties such as hydration, skin elasticity and a reduction in the periorbital wrinkles in 30 days without altering the skin barrier function, sebum, pH, and erythema values. A significant skin moisturizing effect was detected while the skin barrier function was preserved. The selected natural ingredient combined with the designed formulation and the optimized preparation method has led to a final product that satisfies the physico-chemical and technological requirements underlying the safety of use and the formulative stability over time. With no negative skin reactions and highly significant effects on skin elasticity, wrinkles, and moisturization, the CSE-based emulsion achieved very satisfying outcomes representing a promising functional formulation for skin care.

## 1. Introduction

The skin interacts daily with environmental factors and its appearance reflects the well-being of the organism and can affect the social life of an individual. Chronic exposure, especially of the face, to sunlight and pollution or lifestyle habits (smoking, diet) accelerates the aging process (photoaging) that the skin undergoes physiologically (chronoaging). Substantial changes in aged skin affect the epidermis and dermis, such as irregular thickening, uneven pigmentation, decrease in hyaluronic acid and collagen, accumulation of abnormal and highly thickened elastic fibers (actinic elastosis). All these alterations are responsible for the loss of biomechanical properties of the skin, mainly viscoelasticity, the relaxation of its texture, and the formation of solar lentigo and wrinkles [1]. The pathophysiological mechanisms involved in photoaging are complex, and the damage caused by reactive oxygen species (ROS) to cellular structures (nuclear and mitochondrial DNA, membranes, and proteins) are widely recognized as the cause of the destabilization of normal cell functions (signaling, transcription factors, and the expression of certain genes) or cell death. The skin naturally produces enzymatic (catalase, glutathione peroxidase, and superoxide dismutase) and non-enzymatic (such as α-tocopherol, ubiquinone, ascorbate, and glutathione) antioxidants to protect itself from ROS. In the case of external insults, however, the internal defense mechanisms are unable to counteract the overproduction of ROS, and the risk of oxidative stress increases. The topical application of exogenous antioxidants can restore the antioxidant/oxidant balance and prevents ROS injuries [2].

Among exogenous antioxidants, natural derivatives, such as carotenoids and polyphenols are effective, widely present in plants, and potentially induce fewer side effects than synthetic ones [3,4]. There is great interest among researchers and industry in the recovery of antioxidant molecules from waste and by-products of the agri-food chains [5].

Our previous research [6,7] has proved that flavonoid-rich extracts derived from chestnut spiny bur (chestnut harvest wastes) have interesting antioxidant and antifungal properties. The studies support the recycling and the upgrading of this neglected and low-cost material for different applications, such as a natural preservative for vegetable foods or in active packaging. Other authors [8,9,10] have suggested exploiting the antioxidant power of chestnut spiny burs or by-products, such as shell and leaves, for topical skin application.

To carry out its action on the skin, any vegetable extract needs a topical delivery system. The formulation should be able to disperse the extract well, preserve its effectiveness and stability over time, and release it during application at the target site of action [11]. On the other hand, the incorporation of natural active ingredients represents a formulation challenge as they could alter the final product physico-chemical and organoleptic (smell and color) characteristics, influencing the consumer preference. These critical issues, as well as the potential occurrence of adverse reactions, can be overcome by using chemically and biologically standardized phytoextracts as raw materials and developing suitable multicomponent formulations [12]. In any case, in the research and development phase, the physico-chemical and functional stability of new ingredients and final products must be verified. Similarly, it is crucial to estimate the in vitro release of actives and their permeation through synthetic membranes mimicking the skin structure [13].

Oil in water (O/W) emulsions (creams) are widely used for pharmaceutical and cosmetic applications [4]. Due to their biphasic nature, emulsions are able to incorporate and stabilize vegetable ingredients composed of a mixture of molecules with different polarities and solubility [5]. Moreover, creams are generally characterized by ease of application, pleasant skin feeling, and desiderate aesthetic by users [4].

Many formulations (dermo-cosmetics), designed to improve the skin appearance, are recommended by dermatologists to improve healthy skin and as adjuvants in pharmacological treatments of some skin disorders, such as acne, psoriasis, and dermatitis [14]. A topical skincare product applied daily on the face, which is exposed to aggressive factors, can prevent, delay the onset, or treat cutaneous alterations correlated with premature skin aging (photoaging), such as dryness, thinning, decreased elasticity and firmness, hyperpigmentation, and wrinkles [15]. Furthermore, the incorporation of exogenous natural antioxidants (such as polyphenols) represents an added value of the formulation, as many early skin aging signs arise from ultraviolet (UV)-induced overproduction of free radicals and oxidative stress damage [1,4,16].

The European regulation governing the production and marketing of cosmetics supports the dermatological application of many products by imposing safety assessments for raw materials and finished products [17]. Today it is possible to evaluate the benefits of a cosmetic product with non-invasive and scientifically validated techniques that enable safety and efficacy tests on healthy human volunteers through well-designed clinical protocols [18]. For cosmetic and dermatological research purposes or in medical practice, the mechanical properties (elasticity and firmness) and other parameters of the skin layers (such as hydration, transepithelial water loss, sebum, and melanin level) can be quantitatively evaluated by the specific probes of the Cutometer MultiProbe Adapter (MPA) 580 modular system. The effectiveness of a new product on wrinkles’ profile, depth, and the surface can be examined through the fringe projection and 3D images of the skin topography acquired, without any contact, by an innovative device (DermaTop) [19,20,21].

Finally, in the development of a new formulation or natural ingredient for daily skin care, together with instrumental data, it is essential to evaluate the perception of the performance of the product by the consumer, its acceptability, and sensory satisfaction during application. This evaluation can be carried out through self-evaluation questionnaires completed by the volunteers during the experimentation also to describe the perceived changes in skin well-being related to the use of the new product [21,22].

Based on all the above considerations, in the present research, a polyphenol-rich antioxidant extract (CSE) obtained from the chestnut spiny burs was loaded in a new developed multicomponent O/W cosmetic emulsion (B). The immediate (24 h after the preparation at 25 °C) and long term (up to 6 months after the preparation, storage conditions; 4, 25, and 40 °C) physico-chemical stability and rheological behavior was evaluated, with respect to the blank emulsion (A, without CSE). The morphology of the internal structure of the emulsion, the homogeneity of CSE distribution within the emulsion network, the emulsion rheological behaviour, and in vitro skin permeation of CSE through a Strat-M^®^ membrane were investigated. A clinical testing protocol was designed and applied to evaluate the safety of use (by patch test) and effectiveness in improving skin parameters of the developed formulations. Finally, a self-assessment questionnaire was organized to evaluate the perceived quality and sensory appreciation of the volunteers.

## 2. Materials and Methods

### 2.1. Chemicals and Reagents

C_12_-_15_ Alkyl Benzoate (Tegosoft TN), Glyceryl Stearate, PEG-100 Stearate (glyceryl stearate se), Caprylic-Capric Trigliceride (Tegosoft CT), Olea Europea (organic olive oil food/cosm), Glycerin (dermorganics glycerin), Cetearyl Alcohol (50/50 Cetylstearyl Alcohol), Decyl Oleate (Tegosoft DO) were purchased from ACEF Spa (Fiorenzuola D’Arda, PC, Italy). Simmondsia Chinensis Oil (jojoba oil bio cosmos), Dimethicone (Abil 350), Aluminum Starch Octenylsuccinate (Dry Flow PC), Olive Oil Unsaponifiables (Olivene, Squalene Veget, Insap. Olivo), Acrylates/C_10-30_ Alkyl Acrylate Crosspolymer (Carbomer Ultrez 10), Xanthan Gum (xantan gum transparent 80 mesh), Disodium EDTA (Disodium EDTA), l-arginine (l-Arginine-USP), Imidazolidinyl Urea (Kemipur 100) were supplied by ACEF Spa (Fiorenzuola D’Arda, PC, Italy). Phenoxyethanol, Benzoic Acid, Dehydroacetic Acid, Ethylexylglycerin (Euxyl^®^ K 701) was provided by Schülke & Mayr GmbH (Vienna, Austria). The radical 1,1-diphenyl-2-picrilhydrazilic (DPPH), gallic acid, Folin-Ciocalteu reagent, quercetin-3-O-β-d-glucoside, methanol (MeOH) HPLC-grade, MeOH for analysis (MeOH Analar Normapur, Reag. PH. Eur), formic acid (HCOOH), ethanol (EtOH) for analysis (EtOH Reag. Ph. Eur), MTT ([3-(4,5-dimethylthiazol-2-yl)-2,5-diphenyl-tetrazolium-bromide]) were purchased from Sigma-Aldrich (Milan, Lombardy, Italy). The HPLC-grade (18 mΩ) water was prepared with a Milli-Q_50_ filtration system (Millipore Corp., Bedford, MA, USA). Spontaneously immortalized keratinocyte cells (HaCaT) and all reagents and supplements for cell culture were obtained from Gibco Life Technology Corp. (Thermo Fischer Scientific, Weil am Rhein, Germany).

### 2.2. Chestnut Waste Material, Extract Preparation, and Free Radical Scavenging Activity Evaluation

The chestnut spiny burs were collected in October in a chestnut grove (Montella, Avellino, Italy), air-dried and crushed by a blade mill (Grindomix RM 100, Retsch, Bergamo, Italy).

The chestnut burs hydroalcoholic extract was prepared as reported by Esposito et al. (2019) [6]. Briefly, a mixture of aqueous ethanol (50%, *v*/*v*) at 45 °C, was added to a sample (50 g) of dried chestnut burs. The mixture was homogenized by Ultra-Turrax (IKA ULTRA-TURRAX T25 digital, IKA-Werke GmbH & Co. KG, Staufen, Germany), at 10,000 rpm for 8 min, and left for 30 min under magnetic stirring. After filtration through a 45 μm mesh sieve and centrifugation (by Thermo Electron/ALC PK120-V1, Thermo Fisher Scientific, Waltham, MA, USA) for 5 min at 5000 rpm, the liquid extract was processed in a rotavapor (Heidolph Hei-VAP Value Digital Rotary Evaporator, Heidolph Instruments GmbH & Co. KG, Schwabach, Germany) to remove the ethanol. Subsequently, the extract was freeze-dried (Alpha 1–2 LD freeze dryer, Martin Christ, Osterode am Harz, Germany) to remove the residual aqueous portion until a dry extract (CSE) was obtained. The stable 1,1-diphenyl-2-picrylhydrazyl (DPPH^•^) radical was employed to determine the antiradical activity of CSE using the procedures previously described [23,24].

### 2.3. In Vitro Cytotoxic Activity of CSE

Spontaneously immortalized keratinocyte cells (HaCaT) were grown in Dulbecco’s modified Eagle’s medium (DMEM), enriched with 100 mg/L of streptomycin, 10% fetal bovine serum (FBS), and penicillin 100 U/mL, at 37 °C, in a humidified atmosphere at 5% CO_2_. CSE was solubilized in DMSO (dimethyl sulfoxide) to obtain the stock solution (5 mg/mL), from which appropriate dilutions (0.1–3.0 mg/mL) were prepared in a culture medium to not exceed 0.15% (*v*/*v*) of DMSO in all experiments. The cells (1 × 10^4^ cells/well), after 24 h of growth at 37 °C, cultured in 96-well plates, with only the vehicle (DMSO) or with different concentrations of CSE, were mixed with MTT ([3-(4,5-dimethylthiazol-2-yl)-2,5-diphenyl-tetrazolium-bromide]) to obtain 0.5 mg/mL, final concentration. The plates were incubated for 4 h at 37 °C and the formazan product was solubilized as previously reported [24]. The results were obtained by measuring the optical density (OD) of each well at 570 nm (LabSystems, Vienna, VA, USA). The data were normalized with respect to the control and were expressed as cell survival percentage (% viability) [25]. The extract can be classified as not cytotoxic when the cell viability is equal to or greater than 60%.

### 2.4. O/W Emulsion Formulation

The formulation procedure, to obtain A and B emulsions (Table 1), involved heating the two separate oil and water phases at 75 ± 1 °C (Table 1), under magnetic stirring. Once the dissolution of all ingredients for both phases was reached, the oily phase to the aqueous one was added homogenizing for 10 min at 8000 rpm (by Silverson model SL2T, CRAMI Group Srl, Milano, Italy). Gentle manual mixing with a spatula was maintained until cooling. To produce the CSE-loaded emulsion (B, Table 1), the extract (Table 1, Phase C) was previously solubilized in a glycerin:water solution (6.25% *w*/*w*, 4.5 mL) and added to the emulsion during manual mixing once the temperature of 45 ± 1 °C was reached. Preservatives (Table 1, Phase D) and Phase E (Table 1) were added at 35 ± 1 °C. At the end of the emulsification process, the pH was adjusted up to 5.5 using l-arginine as a buffering agent (Table 1, Phase F).

### 2.5. Morphological and Dimensional Analysis

The microstructure analysis of the blank (A) and CSE-loaded (B) emulsions was carried out by means of a fluorescence microscope (Zeiss Axiophot, Carl Zeiss Vision, München-Hallbergmoos, Germany) using a 40 × 1.4 NA project Apochromat objective, oil immersion. The microscope is equipped with a set of three filters housed in the fluorescence filter cube. A DAPI (4,6-diamidino-2-phenyl-diol) was selected as excitation filter and darkfield as oil immersion condenser. Before observation under the microscope, the samples (A and B) were dried on a slide under nitrogen flow.

The size distribution of the oil droplets was detected by liquid module laser scattering diffraction (LS 13 230 Particle Volume Module Plus Beckman Coulter Inc., Brea, CA, USA) [26]. The formulations were diluted in deionized water; a few drops of each sample were poured into the analysis cell until reaching the optimal darkening between 8% and 12%. The volume percentage of droplets dimensional distribution was elaborated with a specific software using the Fraunhofer mathematical model. The analyses were carried out in triplicate: the results were expressed as d_50_ (average diameter) and as span [(d_90_–d_10_)/d_50_], a parameter that takes into account the polydispersity of the droplet size distribution in the dispersed phase; the higher the span value, the greater the inhomogeneity of the size distribution.

### 2.6. Emulsions Physico-Chemical Stability Test

The immediate stability, (T_0_), of blank (A) and CSE-loaded (B) emulsion was assessed 24 h after the preparation, to give the emulsion time to stabilize, for evaluation of the physico-chemical characteristics (pH, phase separation, rheological behavior, and CSE content).

For long-term stability tests, the formulations were stored at different temperatures (4 ± 0.3 °C, 25 ± 0.3 °C, ± 58% relative humidity, and 45 ± 0.3 °C, ± 75% relative humidity) [27], in a climate chamber (Climatic and Thermostatic Chamber, Angelantoni Life Science s.r.l., Perugia, Italy). At the established times (15 days, 1, 3, and 6 months), the samples were taken from the different storage conditions, left to equilibrate at room temperature, and subjected to the same controls performed after 24 h by the preparation.

#### 2.6.1. pH Measurements

The pH control was performed at 25 ± 0.3 °C. The pH value was given by the average of three measurements made with a Mettler Toledo Seven Easy digital pH meter (Mettler Toledo, OH, USA), after a three-point calibration procedure (buffer solutions 4.04, 7.00, and 9.21).

#### 2.6.2. Centrifugation Assay

The centrifugal stability of A and B emulsions was tested on 10 g of product placed in graduated test tubes and centrifuged (Labofuge 200 Centrifuge, Thermo Scientific, Waltham, MA, USA) at 4000 rpm for 30 min. When there was any phase separation or instability events (flocculation, coalescence, creaming, and breaking) further centrifugation was carried out on the same sample at 5300 rpm for 15 min.

#### 2.6.3. Rheological Behaviour Analysis

To investigate the rheological behavior of A and B, a Rheometer MCR102 (Anton Paar, Physica, Austria) with cone–plate geometry (CP50-1-SN53715, diameter 49.980 mm, cone angle 0.989°, cone truncation 99 µm) and RheoCompass™ Software was employed, in continuous and oscillatory flow conditions. Flow behavior was determined in logarithmic mode (by increasing shear rates from 0.01 to 1000 s^−1^), applied for 30 s with 21 points per decade. The flow curves were obtained by plotting the viscosity (in mPa s) as a function of shear rate (in·s^−1^). Amplitude sweep tests were performed at a fixed frequency of 10 1/s and increasing strain ranging from 0.01 to 100% (log mode, 21 points/decade). The curves were obtained by plotting the storage modulus (G’, Pa) and loss modulus (G″, Pa), versus share strain (γ, %) to analyze the linear viscoelastic region (LVR). A frequency sweep test was carried out at a constant strain of 0.5%, increasing angular frequency (ω) from 0.1 to 100 rad·s^−1^. All measurements were performed twice on two different batches, at 22.0 ± 1.0 °C.

#### 2.6.4. Evaluation of the Chemical and Free Radical Scavenging Stability of CSE in B Emulsion

The recovery of CSE from the formulation B (kept at 25 ± 0.1 °C) was performed 24 h and 6 months after the preparation, according to the following procedure [25]: 1.0 g of B was extracted with 10 mL of a hydroalcoholic mixture (EtOH:H_2_O, 1:1, *v*/*v*), subjected for a few seconds to manual stirring in falcon, and centrifuged at 5300 rpm for 10 min at room temperature.

The supernatant was recovered, dried by rotavapor (Heidolph Hei-VAP Value Digital Rotary Evaporator, Heidolph Instruments GmbH & Co. KG, Germany), and analyzed by the Folin–Ciocalteu test [28]. The content of recovered extract (actual extract content, AEC) was calculated by Equation (1) on the basis of the total phenol content (TPC, expressed as gallic acid equivalent, GAE µg/mg extract) of the supernatant (TPC_emuls_) and the TPC of the raw extract (TPC_CSE_):AEC = TPC_emuls_/TPC_CSE_(1)
and from this, the recovery percentage (RE%) was obtained by Equation (2):RE% = AEC/TEC(2)
where TEC is the theoretical extract content in the tested emulsion.

To 1.5 mL of DPPH solution were added 37.5 µL of the recovered supernatant and the mixture was kept in the dark for 10 min at 25 ± 1 °C [25]. The change in absorbance was monitored at λ = 517 nm (Thermo Evolution 201 UV–visible spectrophotometer, Thermo Fisher Scientific Italia, Milan, Italy) with respect to the blank, consisting of an equal volume of solvent (EtOH:H_2_O, 1:1). Tests were performed in triplicate. The following formula (Equation (3)) was used to determine the radical scavenging activity (RSA):% RSA = (A_0_ − A_s_)/A_0_ × 100(3)
where A_0_ is the absorbance of a control solution prepared without a sample, and A_S_ is the mean absorbance value of the solution prepared with the sample.

### 2.7. In Vitro Permeation Studies of CSE from O/W Emulsion

The permeation of CSE through a synthetic membrane and its release from formulation B was determined by Franz vertical diffusion cells (Hanson Research Corporation, Chatsworth, CA, USA). Strat-M^®^ (25 mm, Merck, Darmstadt, Germany), despite being synthetic, is very useful in predicting the diffusive behavior of the phytoextract through human skin as it has a good morphological affinity with the skin [13]. It is a multilayer membrane of polyethersulfone, which has a very compact surface, with increasing porosity as it moves along the various layers. The permeation area is equal to 1.7671 cm^2^ and the liquid medium used in the acceptor compartment (volume of 7 mL) under magnetic stirring at 170 rpm was aqueous ethanol (10% *v*/*v*).

1 g of CSE-based cream, with an extract content of 3 mg (0.3% *w*/*w*) was applied evenly to the surface of the Strat-M^®^ membrane, covered with a clip, and fixed with Parafilm^®^. The high amount loaded on the donor compartment was due to the impossibility of uniformly covering the membrane with a smaller quantity ensuring the application of a homogeneous distribution. At the same time intervals, 200 µL aliquots were taken from the acceptor compartment and immediately replaced with an equal volume of fresh solvent. The samples thus collected were investigated by HPLC-DAD for the determination of quercetin-3-O-β-d-glucoside, used as a marker of CSE [6]. The permeated CSE value per area (Q) for each time interval was calculated according to the following Equation (4):(4)$$Q\cdot \left(\frac{mg}{cm^{2}}\right)=\frac{V_{R}\times C_{N}+\sum_{i=n}^{n-1}V_{P}\times C_{i}}{A}$$
where *V_R_* is the receiver volume; *C_N_* is active concentration in the receiver at the time *n*; *V_P_* is the volume of the removed sample; and *C_i_* is active concentration in the receiver at the time *n* − 1.

As a control sample, an aliquot of 3 mg of freeze-dried CSE solubilized in a 50% ethanol solution was loaded on the membrane. At predetermined times (5, 10, 15, 20, 30, 40, 50, 60, 90, and 120 min), samples of 200 µL were taken from the acceptor compartment. The fractions thus collected were stored in Eppendorf at 4 ± 0.1 °C, and subsequently injected into the HPLC.

Results were also expressed as percentage of dose permeated at the end of the test (2 h) with respect to the total amount of extract (3 mg) loaded on the donor compartment.

An Agilent 1100 series system including a Model G 1312 pump, a Rheodyne Model G-1322A loop (20 μL), a DAD G-1315 A detector (Agilent Technologies, Waldbronn, Germany), and an Agilent integrator, were used to perform the quantitative HPLC analysis. Detection by diode array was set at 278 nm. Chromatographic separations were achieved with a Luna C_18_ (240 mm × 4.6 mm, 5 μm, Phenomenex) column, using water (solvent A) and methanol (solvent B), both containing 0.1% (*v*/*v*) formic acid, as mobile phase. The analysis was performed at a flow rate of 1 mL/min, injection volume of 20 µL and using the following solvent gradient: 0→5 min, 5→22% B; 5→8 min, 22→35% B; 8→13 min, 35→50% B; 13→23 min, 50→70% B, 23→33 min, 100% B. To identify the peak associated with quercetin-3-O-β-d-glucoside its retention time was studied and confirmed by co-injection with the standard compound. The standard curve was obtained using three concentration levels in the range 0.025–1.0 mg/mL (LOD = 0.001 mg/mL) [29] and analyzed using the linear correlation between the concentration and the peak area (regression equation was y = 16,560 × −341.27, R^2^ = 0.9984 where y is the peak area and x the concentration).

### 2.8. Clinical Trial Testing

#### 2.8.1. Research Ethics Committee Approval

A single-blind, placebo-controlled, monocentric study was designed to evaluate the skin tolerability and the skin performances of CSE-loaded and blank emulsion. The clinical study was carried out according to the declaration of Helsinki [16,20,30,31] for medical research involving human healthy subjects. The study protocol was evaluated and approved by the Ethics Committee of San Giuseppe Moscati Hospital, Avellino, Campania (Italy) (Project identification code, CECN/1333, session of 28 October 2020).

#### 2.8.2. Volunteer Recruitment

For the study, 20 female volunteers aged between 32 and 54 years (average age 40 years) with Fitzpatrick phototype from II to IV, and relative frequency in the use of cosmetic products were enrolled. The dermatologist provided clinical observation of the volunteers’ skin and verified with an interview the compliance with the following inclusion criteria for each of them:State of pregnancy or lactation;Allergy or skin reactivity to cosmetic ingredients;Skin disorders (dermographism, seborrhea, dermatitis, Herpes, Pityriasis Versicolor, or psoriasis);Skin pigmentation changes (vitiligo, chloasma, or chronic Lupus erythematosus);Surgical interventions in the last year in the areas to be tested;Injuries, spots, or marks (sunburns, scars, tattoos) in the experimental skin area;Systemic or topical pharmacological treatments (NSAIDs, antihistamines, or cortisone) in the 10 days before the start of the trial;Invasive aesthetic treatments (peeling, laser, biostimulation, filler, botulinum toxin, or injections based on hyaluronic acid and vitamins) in the 2 months before the study;Non-invasive aesthetic treatments (scrubs, facial cleansing, or radiofrequency) in the month preceding the study;Excessive exposure to sunlight or UVA rays in the month before the study;Application on the skin area under examination of cosmetic products similar to the one tested in the 10 days before the start of the experiment.

The volunteers were required to promptly notify the dermatologist in the event of skin reactions or discomfort sensations during the application of the products.

Moreover, the selected volunteers were asked: not to apply other products similar to the tested one, keep up hygiene and/or makeup habits, and have full compliance with the use conditions of the tested products.

Each volunteer signed the informed consent form accepting restrictions, test modalities, and responsibility for the statements made to the investigators.

#### 2.8.3. Patch Test

The tests were carried out under the supervision of a dermatologist at the hospital San Giuseppe Moscati of Avellino. 10 µL of emulsion A and B were separately loaded on a Finn Chamber (Curatest^®^, Lohmann e Rauscher, Rengsdorf, Germania) device applied on the right (A) and left (B) outer forearm of each volunteer. The devices were fixed at the application site with adhesive patches and removed after 48 h. The clinical assessment of skin irritation was made by dermatologist 1 h and 24 h after the removal of the Finn Chamber, according to the following score scale: absence of erythema/edema (0); slight (hardly visible) erythema/edema (1); light (clearly visible) erythema/edema (2); moderate erythema/edema (3); severe erythema/edema (4). The sum of erythema and edema scores is defined as the irritation index. The average of the 20 irritation indices was calculated and the emulsions A and B were classified according to an amended Draize classification (Table 2) [30].

#### 2.8.4. Efficacy Protocol

The volunteers were instructed on the application modalities: blank emulsion (A) on the right side of the face and CSE-loaded emulsion (B) on the left side, twice a day, morning and evening, for 30 consecutive days, gently massaging a useful amount of product until completely absorbed. All instrumental tests were carried out in the same experimental center, at the established times (t_0_ = before applying the products, t_15_ = 15 days, and t_30_ = 30 days of application), at an average temperature of 20 ± 2 °C and humidity of 55 ± 2%. Before each measurement, the volunteer was subjected to a 15-min adaptation to laboratory conditions. Non-invasive measurements of the skin parameters, hydration, elasticity, softness/stiffness, transepidermal water loss (TEWL), pH, sebum, and pigmentation were carried out on the right (emulsion A) and left (emulsion B) cheekbones or cheeks (elasticity) using the probes of the Cutometer^®^ dual Multiprobe Adapter System (MPA) 580 (Courage and Khazaka, Electronic GmbH, Cologne, Germany). The skin topography (wrinkles depth measurement) was investigated by the DermaTOP system (DermaTOP^®^, Eotech, Marcoussis, France). Four measurements were performed for each skin parameter at different detection times.

##### Skin Hydration

The assessment of the *stratum corneum* hydration was performed by the Corneometer^®^ 825 probe, and it is based on the capacitance method. Pressing on the skin surface (depth of 10–20 μm) for a few seconds, the probe head, presenting a precision capacitor, measures the dielectric constant of the stratum corneum. The large difference between the relative dielectric constant of water (εr = 81 C^2^/Nm^2^) and other substances of the skin (εr < 7C^2^/Nm^2^) is the basis of the measurement. Changes in the dielectric properties, depending on the surface water content, are registered when the first layer of the skin comes into the scattered field of the condenser, and are converted in arbitrary Corneometer^®^ units (higher values correspond to better hydrated skin) [19,32,33,34].

##### Skin Elasticity

The viscoelastic properties of the facial skin were evaluated by the Cutometer^®^ Dual MPA 580 probe using a suction method. A negative pressure setting to 450 mbar is created by the device, to deform the skin, pulled into the opening (2 mm Ø) of the probe. After 3 s the skin is released and after a relaxation time of 3 s, it is pulled again (the cycle was repeated three times). The device outputs live strain–time curves [deformation (mm)/time (s)] (Figure 1) allowing measurement of the skin resistance to suction and the ability to return to its original shape and position. Several associated parameters related to skin elasticity were derived by the Cutometer^®^ software. Among them, the parameters R2, R5, and R7 were considered. R2, also called gross elasticity, is the expression of visco-elasticity (the ratio of the skin resistance to the suction phase and ability to return during the relaxation step, Ua/Uf); R5 defines the net elasticity (the elastic portion of the suction phase versus the elastic portion of the relaxation phase, Ur/Ue); R7 is the ratio of elastic recovery to the total deformation (Ur/Uf). The closer the R2, R5, and R7 values are to 1 (100%) the more elastic the skin [32,33,35].

##### Skin Softness/Stiffness

The cylindrical indenter of the probe Indentometer^®^ IDM 800 pushes the skin, and the penetration depth is evaluated in mm (0–3 mm). The more firmness/stiffness of the skin, the less displacement (in mm) by the probe [36].

##### Transepidermal Water Loss (TEWL)

The TEWL was assessed by the TEWAMETER^®^ TM 300 probe. The device allows an “open chamber” measurement. It is made up of an empty cylinder equipped with temperature and relative humidity sensors that continuously measure the evaporation rate of water (in·g/h/m^2^) from the skin [34].

##### Skin pH

The pH of the skin surface was evaluated with the probe Skin-pH-Meter^®^ PH 905 after calibration. The measurement takes place in 1 s avoiding occlusion effects. The resulting pH value is expressed to one decimal place [37,38].

##### Skin Sebum

The skin surface sebum level was determined with the Sebumeter^®^ SM 815 probe. The evaluation is based on the photometry of the grease spots. The mat tape (thickness of 0.1 mm) of the probe, placed on the area of interest (64 mm^2^), becomes transparent according to the sebum amount. The tape transparency is measured by a photocell of the device and the light transmission is related to the sebum content. A microprocessor calculates the result in units on a 0–350 units scale [37,38].

##### Melanin Index

The probe Mexameter^®^ MX 18 was used to measure the level of melanin and hemoglobin (erythema), the two components mainly responsible for the color of the skin. The probe emits light at defined wavelengths (λ 568, 660, and 880 nm), corresponding to the spectral absorption peaks of melanin (660 and 880 nm) and hemoglobin (568 nm). A receiver measures the reflected light from the skin. Knowing the amount of light emitted by the probe, the amount absorbed by the skin is determined. The results were reported as arbitrary Mexameter^®^ units (0–999 for melanin and erythema) [37,38].

##### Wrinkle and Fine Line Measurements

The DermaTOP, based on a patented fringe projection unit obtained by an electroluminescent diode projector (LED), determines objective parameters to evaluate skin and morphology changes due to age, treatment, or product application. The produced fringes are deformed proportionally to the relief. The images are recorded by a charge-coupled device (CCD) camera and analyzed by AEVA software to reconstitute the three-dimensional profile of the area under examination in few seconds. The DermaTOP system was used to acquire 3D information on skin topography, such as depth, volume, and roughness of crow’s feet and fine lines in the periorbital area. In a dark room, the camera was positioned (without contact) on an area (20 × 20 mm) of the right and left temple to capture 50 crow’s feet wrinkles profiles. A 50 mm depth-of-field lens was used. From the captured 3D images, wrinkle parameters (Ra, Rt, and Rz) were calculated. Ra represents an arithmetical average of wrinkle parameters, the average deviation of the profile from the mean line (arithmetic mean of the absolute values of the point’s heights), for wrinkle assessments, representing the finer skin structure. Rt is defined as the maximum peak to valley height of the profile in the assessment, length, and Rz is the rough structure, the average maximum profile height difference. A reduction in the depth of wrinkles is indicated by a reduction in these parameters [29,39].

### 2.9. Self-Assessment Questionnaires of Creams A and B

Each volunteer answered questions regarding the subjective evaluation of the effectiveness and cosmetic characteristics of the two products. At t_0_ (before the product application) and at the end of the clinical testing (t_30_) the subjects expressed a judgment on their facial skin characteristics in relation to the state of smoothness, elasticity, softness, brightness, and hydration. The responses obtained at t_0_ and t_30_ were compared. Moreover, after 30 days of treatment, the volunteers expressed their opinions regarding the following cosmetic properties of the tested products: easy of application; absorption; and overall acceptability according to a four-point descriptive scale: low, sufficient, good, or excellent. The results were expressed as a percentage.

### 2.10. Statistical Analyses

All the data were analyzed by using GraphPad Prism version 7.00 for Windows. The data have been reported as mean ± SD. The measurements were analyzed by one-way analysis of variance (ANOVA) followed by Tukey’s multiple comparisons test (*p* ≤ 0.05). The measurements obtained by in vivo studies were analyzed by a two-way RM (repeated measures) ANOVA (analysis of variance), followed by a Bonferroni adjustment to determine the statistically significant differences between the measurement time and formulation (*p* ≤ 0.05).

## 3. Results and Discussion

### 3.1. Chestnut Burs Free Radical Scavenging Extract Preparation and Cytotoxic Activity Evaluation

In our previous study [6], a dry extract, CSE, was prepared from chestnut burs by dynamic maceration (with ethanol:water 1:1, *v*/*v*) preceded by homogenization. CSE presented a high total polyphenol content expressed as gallic acid equivalent (TPC, 20.60 GAEg/100 g extract). In our previous research [6] the extract chemical profile was investigated, showing that CSE contains two main classes of polyphenols, namely, hydrolysable tannins and flavonoids. Among the CSE polyphenols, in the present study, quercetin 3-O-β-d-glucopyranoside [6] was selected as the extract marker. CSE showed good scavenging capacity against the DPPH radical (EC_50_ 24.94 ± 0.46 µg/mL, EC_50_ of positive control (gallic acid) = 1.23 ± 0.15 µg/mL). The topical skin application of exogenous natural antioxidants seems to be an effective strategy to prevent or counteract the damage and cell aging induced by free radicals on skin that is daily in direct contact with UV radiation and pollutants [9,10,40]. As the aim of the present research was to incorporate the polyphenol-rich CSE in a skin care product, the cytotoxicity of the extract on spontaneously immortalized human keratinocytes (HaCaT) cells was evaluated. The results of the MTT ([3-(4,5-dimethylthiazol-2-yl)-2,5-diphenyl-tetrazolium-bromide]) test showed that, after 24 h of incubation, CSE did not reduce the cell viability of keratinocytes up to a concentration of 3 mg/mL (Figure 2).

### 3.2. O/W Emulsion Formulation

Due to their biphasic (hydrophilic and hydrophobic) nature, emulsions are semisolid formulations suitable for encapsulating a complex ingredient, such as a hydroalcoholic phytoextract that contains a mixture of molecules with different polarity and solubility [41]. A new oil/water (O/W) emulsion was designed and developed as a topical dosage form to deliver CSE to the stratum corneum. A careful choice of ingredients (Table 1) approved for cosmetic use (according to EC Regulation no. 1223/2009) [17] was made.

To obtain a personal care product with high skin compatibility, most of the selected ingredients were of vegetable origin (emulsifying and viscosity controlling agents and emollients in the oily phase) or naturally occurring in the skin (such as the amino acid l-Arginine). Some of the employed raw materials are able to improve the dispersion of the CSE ingredient (such as C_12_–C_15_ Alkyl Benzoate and Glycerin), and to control the texture and rheological properties of the formulation. Furthermore, the ingredients, such as olive oil (Olea Europea oil), unsaponifiable olive oil (Olea Europea oil unsaponificables), jojoba oil (Simmondsia Chinensis oil), and decyl oleate can give qualities, such as hydration, emollience, and antioxidant effects acting in synergy with the new extract (CSE) to exert the functional performance.

Formulation A was optimized in terms of either ingredient percentages, within the range of use indicated by the manufacturers and according to EC Regulation no. 1223/2009, and preparation method (time and speed of phases homogenization).

CSE was loaded into the emulsion (B, Table 1) during the cooling phase (at 45 °C) to avoid chemical degradation of the heat-sensitive polyphenols. The concentration (0.3%, *w*/*w*) was established based on the effective and non-cytotoxic concentration. The incorporation of a higher percentage of CSE was tested (up to 0.5%, *w*/*w*), but the obtained formulation, despite still being stable (no separation phase after centrifugal test), presented organoleptic characteristics (dark brown color and very intense odor) not totally compatible with a sensory appreciation by the final consumer.

### 3.3. Morphological Analysis and Dimensional Analysis

To verify the distribution of the active extract CSE into the O/W emulsion system a fluorescence microscopy analysis was assessed (Figure 3).

Both the blank (Figure 3a) and CSE-based (Figure 3b) emulsion are well structured; from the photomicrographs, a dense network of micrometric oil droplets (droplet size d_50_ 2.8–2.9 µm) appears. Due to the presence of several lipids forming an oily phase of emulsions, the oil droplets dimensional distribution within the aqueous phase was not completely homogenous for both formulations, as graphically demonstrated by the bimodal curves (Figure 3c,d), with a polydispersity index of about five for each formulation. Nevertheless, the oil droplets were well surrounded by the water phase that is visible in blue, for the blank emulsion image (a), and as a pale yellow for the CSE-loaded emulsion image (b). The difference in the color (blue or yellow) is due to the presence of naturally fluorescent molecules within the extract [11,25]. The extract was perfectly dissolved in the aqueous phase of the emulsion and did not interfere with the polymeric network created by the ingredients of the formulation. The relationship between excipients and extract appears to be well balanced and the selected preparation method led to obtaining a well-structured emulsion.

### 3.4. CSE and Emulsions Physico-Chemical Stability Test

The chemical stability of the polyphenols after the preparation process and storage period is a critical point to preserve the antioxidant activity of the raw vegetable extract within the formulation. In addition, polyphenolic compounds can promote phenomena of emulsion instability (phase separation, flocculation, creaming) by precipitation or chemical interactions with other ingredients. The emulsion, as the delivery system, has to allow both their stability and release to the target site [42,43]. For these reasons, in the development of a new formulation it is essential to evaluate the physical-chemical stability of the product as it relates to quality, effectiveness, safety, and acceptance by the consumer. Therefore, the immediate and long-term (up to 6 months after preparation) accelerated stability, under different storage conditions (at 4, 25, and 45 °C) of formulation B (loaded with CSE) was evaluated and compared with that of the blank formulation A.

Storage at different temperatures accelerates the kinetics of oxidation, degradation, and interaction of components, or phase separation. For both formulations, the appearance of instability phenomena after centrifuge tests, pH values, and rheological behavior were investigated. Centrifugal force accelerates sedimentation and surfacing and allows to predict the shelf-life of an emulsion over time. Formulations A and B kept at 4, 25, and 45 °C were considered stable with no phase separation or other signs of instability observed up to 6 months.

The pH of a topical product must fall within the physiological range of the *stratum corneum* (4.1–5.8, except for the eye area) to avoid irritation after application, preserving the integrity of the epidermal barrier and the skin antimicrobial defenses [44,45]. Moreover, pH stability is crucial for the preservative system to maintain its antimicrobial activity, while pH could be also an indicator of microbial pollution if its value changes over time.

Emulsions A and B presented pH values (range 5.43–5.58) compatible with skin application (Table 3). No significant (*p* > 0.05) pH changes were observed during the storage period of A at different temperatures. The addition of CSE did not affect the pH value of the emulsion.

### 3.5. Rheological Assessments

The rheological analysis is a prerequisite to evaluate the internal structure of semi-solid materials, such as emulsions, their firmness, stability, and shelf-life from the production phase, to the packaging, until the application on the skin [46]. Moreover, also the acceptability of a cosmetic product by the final consumer is closely linked to the flow and deformation properties, which can be determined by continuous and oscillatory rheological tests [46]. The flow test was carried out to describe how the material is deformed when subjected to stress. The obtained results (Figure 4) suggested that A and B can be classified as shear-thinning, stable, easily rubbed emulsions, with good spreadability features, and potentially well absorbed through the skin [47].

As matter of fact, formulation A (gray line) showed a reduction in dynamic viscosity (η) value as the shear rate (Ύ) increased (viscosity values range from 315.79 mPa·s to 151.030 mPa·s) (Figure 4a). This behavior is typical of non-Newtonian fluids, such as creams, gels, lotions, and emulsions [5,48]. The profile indicates the semi-flexible molecular structure of A, with a low flow resistance; aggregates are rapidly broken into their elements, the droplets lining up at a high shear rate supports the formulation’s ability to spread on the skin [21]. A similar profile was obtained for B (Figure 4a, black line), in which the presence of CSE did not affect the flow curve (viscosity values range from 311.46 mPa·s to 158.08 mPa·s). Moreover, the same trend was seen in both formulations (Figure 4b,c) even up to 6 months at different storage temperatures (4, 25, and 45 °C).

The emulsion’s ability to form a stable gel-like network and its susceptibility to deformation was investigated by dynamic oscillatory measurements, identifying the linear viscoelastic region (LVR) [21,49]. A strain sweep test was performed to determine the viscoelastic parameters, elastic modulus G′ (elastic response), and viscous modulus G″ (viscous behavior) within LVR.

At low deformation values, the predominance of the elastic (G′) over the viscous (G″) modulus (Figure 5) indicated a stable solid-like behavior of both A and B formulations [50]. As the strain increased, the elastic component (G′) decreased in favor of the viscous component (G″), meaning that the emulsions became progressively fluid-like, and so to flow on the skin more easily [51]. The analysis of LVR (value around 1%) and G′ values (higher than 10 Pa) suggested that the produced formulations were well structured, as also shown by morphological analysis (Figure 3) and classifiable as hard-base creams [21,49]. Moreover, the greater distance of the cross over (G′ = G″) from LVR confirm that the blank and CSE-loaded formulations can be considered stable products [52].

The curves deriving from the frequency sweep test were realized by interpolating G′ and G″ (Figure 6a) or the complex viscosity (*η**; Figure 6b) with angular frequency, at a fixed strain, within LVR. A and B can be considered stable emulsions owning G′ > G″ over the entire measured frequency range (Figure 6a). Moreover, the absence of a crossover in Figure 6a suggested that each emulsion can be considered as non-sticky [46,47]. A linear reduction in log η* was observed (Figure 6b), confirming the gel-like structure and viscoelastic solid behavior. The same profile was observed for A and B stored at different temperatures for up to 6 months (data not shown), predicting the long-term stability [51]. The presence of CSE did not interfere with the intrinsic structure of the formulation, maintaining B an elastic structure over the range of frequency (Figure 6b, black line).

### 3.6. Evaluation of the Chemical and Free Radical Scavenging Stability of CSE in the O/A Emulsion

To evaluate the chemical and functional stability of CSE loaded in the emulsion, an aliquot of B was diluted with aqueous ethanol (1:1, *v*/*v*), the optimal solvent for the recovery of the hydroalcoholic extract from B. The supernatant obtained after centrifugation was dried and analyzed by Folin–Ciocalteu test. The results showed that both 24 h and 6 months after preparation the percentage of CSE recovered (RE%) was 84% (variation < 1%) of the total extract incorporated in the formulation (TEC). The recovered extract was submitted to DPPH testing, and showed a radical scavenging activity (RSA%) of 90.04 ± 1.32%, with no significant (*p* > 0.05) differences after 24 h and 6 months. The constant recovery rate and free radical scavenging activity of CSE can demonstrate the ability of B to preserve the long-term chemical functional stability of polyphenol compounds.

### 3.7. In Vitro Permeation Studies of CSE from O/W Emulsion

The permeation of the active ingredient through the skin is one of the main factors contributing to efficacy; its diffusion depends on the physiological characteristics of the skin but also on the excipient’s formulation and physico-chemical characteristics of the active ingredient itself. The effect of the emulsion composition on the CSE permeation was investigated in vitro using the synthetic membrane Strat-M^®^ that is considered an adequate model for replacing animal or human skin [53]. Strat-M^®^ is a recently engineered multilayer membrane, in which the composition of *stratum corneum*, *epidermis*, and *dermis* is reproduced, without the lot-to-lot variability typical of human skin samples traditionally used to evaluate the transepidermal permeation of pharmaceutical or cosmetic active ingredients.

In vitro tests on Strat-M^®^ synthetic membrane were carried out to assess the impact of the emulsion’s semi-solid base nature on the release of the active ingredient. Generally, emulsion-type formulations, which are multiphase hydrophilic–lipophilic systems, can favor the release of the extract from the formulation also enhancing permeation through membranes of different active ingredients compared to monophasic systems.

As also shown by the Fluorescence image (Figure 3b) the good solubilization of the extract in the emulsion network improved the permeation and potentially enhances diffusivity across the membrane.

The amount of CSE able to cross the membrane (Figure 7) was about of 178.00 µg/cm^2^ every 5 min and it was kept constant during the first hour of the test reaching higher permeation during the second hour (296.50 µg/cm^2^). Based on the EC_50_ value (24.94 µg/mL) determined by the DPPH test, CSE permeation behavior could ensure that the active dose is able to exert the free radical scavenging effect. The obtained results (Figure 7) showed that the incorporation of the extract into the emulsion represents a valid strategy to enhance the permeation of the phytoextract underlining the importance of the soluble state of the extract and its high concentration to ensure activity.

### 3.8. Clinical Trial Testing

#### 3.8.1. Patch Epicutaneous Test

To avoid side effects the assessment of the irritant or allergic hazard becomes crucial before placing a new skincare product on the market [17,30]. The safety of the developed leave-on emulsions A and B, intended to be applied for a long time on facial skin, and composed of several ingredients, including a new plant derivative, was clinically determined by patch test. The results were evaluated by the dermatologist through the visual observation of erythema or edema appearance at the application area (right and left forearm for A and B, respectively). The tested products applied as they are under occlusive conditions (48 h) on the healthy skin of volunteers resulted in a mean index of irritation of 0.0, 1 h, and 24 h after the patch removal. The products can be classified as not irritating (average irritation index from 0.0 to 0.5, Table 2) (if applied to intact human skin).

#### 3.8.2. In Vivo Evaluation of Emulsions Effectiveness

Premature skin aging (photoaging) is an alteration induced by many exogenous insults (sunlight, air pollution, toxins, or lifestyle) which cause changes in the epidermis appearance, as well as loss of structural integrity, and reduction in functionality of the dermal extracellular matrix components (such as collagen and elastin fibres) [1].

To investigate the effectiveness of the developed formulations and to prove a potential different effect between A and B, due to the addition of the new polyphenol-rich CSE ingredient, some biomechanical and biophysical skin parameters were evaluated by non-invasive devices.

Moreover, biophysical parameters such as TEWL, pH, quantity of sebum, and melanin content were evaluated to determine the influence of the new products, A and B, with prolonged application on skin barrier function and integrity.

##### Skin Hydration

Hydration of the *stratum corneum* positively affects the elastic properties and flexibility of the skin. A reduced water level occurs in aging or the skin with impaired barrier function and contributes to the appearance of skin disorders, such as desquamation, itching, stinging, and even psoriasis and eczema [32,33]. The daily routine with moisturizing formulations can cooperate with the endogenous systems, to maintain an adequate level of skin hydration. After 30 days (T_30_) the use of both A and B significantly (*p* < 0.0001 ****) improved the skin hydration level (the mean value of arbitrary corneometer units at T_30_ was 57.80 for A and 56.68 for B), compared to T_0_ (the mean value of arbitrary corneometer units was 33.07 for A and 32.56 for B) (Figure 8). Formulation B allowed to obtain this positive result already after 15 days (T_15_) of application (Figure 8, black boxes) (mean value of arbitrary corneometer units at T_15_ was 42.39 for A and 43.41 for B). Our results show that the extract CSE cooperates with the other ingredients to ensure a moisturizing effect of the formulation already after 15 days of constant application.

##### Skin Elasticity and Skin Softness/Stiffness

The viscoelastic properties of the skin were evaluated through the principle of suction. The measurements were made in the nasolabial area of the cheek, where the sagging of the aged skin is more evident [35]. From the strain–time curves the parameters R2 (gross elasticity), R5 (net elasticity), and R7 (elastic recovery), which are the most suitable to show the reduction in skin mechanical properties, were considered. On the face, R2, R5, and R7 showed a negative correlation with skin aging (the higher the values, close to 1, the more elastic and “younger” the skin) [54]. Reduced firmness also indicates less elastic skin. The probe Indentometer IDM 800^®^ was used to quantify the skin stiffness [36].

Applying B for 30 days (T_30_) significantly improved the elasticity parameters, expressed as mean values of cutometer units (Figure 9a, black boxes). R5 increased from 0.4889 (T_0_) to 0.5960 (T_30_); R2 from 0.6068 (T_0_) to 0.6899 (T_30_); and R7 from 0.3825 (T_0_) to 0.4548 (T_30_). The CSE extract contributes to the effectiveness of B by enriching the emulsion with polyphenols. Polyphenols, in fact, seem able to exert a protective effect against oxidative stress-induced cellular disorders, downregulating the expression of metalloprotease, and reducing degradation or stimulating the deposition of elastic fibers [1,55].

On the contrary, neither A nor B had a significant influence on the skin stiffness (Figure 9b) after 30 days of application.

##### Transepidermal Water Loss (TEWL), Skin pH, Skin Sebum, Melanin and Erythema Index

The diffusion gradient of the water from the outer layer (TEWL) is controlled by a fully functioning stratum corneum; the maintenance of an acid pH (4.6–5.6) is optimal for enzyme activity and keratinocytes cohesion. Moreover, the presence of sebum (a mixture of non-polar lipids) allows the skin to counteract the attack of microorganisms and maintain its integrity [37]. Finally, changes in skin color can indicate local irritant reactions and contact dermatitis [38]. The effect of the formulations on these parameters was investigated to demonstrate the absence of skin alterations due to prolonged use of the products.

Measurements showed that both formulations A and B did not significantly alter TEWL values after 30 days of application, maintaining very good skin conditions (Figure 10a, range 4.5–10.7 g/h/m^2^). Even the sebum (Figure 10b, range 12–25 µg/cm^2^), the acid pH (Figure 10c, range 4.5–5.7), and melanin (Figure 10d, range 97.33–225.33 Mexameter^®^ units) content were not affected.

These data, together with those of the patch test, confirm the safety of the developed formulations.

##### Analyses of Wrinkles’ Appearance by Dermatop Evaluation

The first sign of skin aging is the appearance of facial roughness and wrinkles caused by the breakdown of dermal collagen and elastic fibers, the poor *stratum corneum* hydration, as well as gravity and continuous expressive movements [1]. The investigation of A and B’s effectiveness in reducing the appearance of wrinkles and fine lines in the periorbital region (crow’s feet) was carried out by DermaTop. This innovative no-contact technique of image analysis is employed to support the anti-wrinkle claims of dermal products [29,39]. The Aeva software acquires the images in the same area of interest, at different times of analysis (t_0_ and t_30_ days), and overlaps them, to calculate the roughness parameters (Ra, Rz, and Rt). The application of A and B visually reduced the depth of wrinkles in the periocular area, as indicated by the less red region in the 3D images acquired after 30 days (t_30_) of application compared to t_0_ (Figure 11a,b). However, only the Rt (the maximum relief heigh) parameter significantly decreased after the application of the formulations. Formulation A induced a reduction of Rt from 0.2027 to 0.1676 mm (mean values at T_0_ and T_30_, respectively) (*p* < 0.05 *) and B from 0.2571 to 0.2076 (mean values at T_0_ and T_30_, respectively) (*p* < 0.001 ***) (Figure 12b). The change in the microrelief aspect can be attributed to the moisturizing effect of A and B, applied on the skin with intact barrier function [56]. Moreover, the presence of antioxidant molecules, such as polyphenols, can act by quenching the free radicals and protecting collagen from degradation [4]. The results showed that the multi-ingredient composition, including the functional vegetable extract, of the developed emulsions can improve the skin appearance by reducing wrinkles and roughness.

### 3.9. Self-Assessment Questionnaires of Creams A and B

Subjective consumer assessments are essential in the development of a new product to ensure its success on the market [21,22]. For this reason, at the end of the treatment, the volunteers were questioned on the perceived effectiveness and sensory properties of A and B separately.

The results (Figure 13) were expressed as a percentage of subjects who rated facial skin as more hydrated, bright, soft, elastic, and smooth after 30 days, compared to t_0_. A higher percentage of volunteers (between 75 and 95%) perceived improvements in the state of skin well-being after application of B (Figure 13a, black line), compared to A (Figure 13a, grey line). These improvements were felt already after only one week of use of B. Furthermore, 85% of the volunteers rated as excellent the cosmetic characteristics, ease of application, and skin absorption of B (Figure 13b). In conclusion, B met with the approval of the volunteers, both for its effects and feel on the skin, and it was used in the volunteers’ daily beauty routines, even after the end of the experiment.

## 4. Conclusions

This work has proved the potential application of a new stable oil-in-water emulsion enriched with the antioxidant hydroalcoholic chestnut spiny burs extract (CSE), as an active skin care formulation. The selected natural ingredient combined with the designed multi-component formulation and an optimized preparation method has led to obtaining a final product that satisfies the physico-chemical and technological requirements underlying the safety of use and the formulative stability over time.

Clinical studies have shown that enrichment with the active CSE ingredient does not alter the skin tolerability of the basic formulation and increases its functional effectiveness after daily application on the face. The presence of the extract, just after 15 days, improves skin hydration and reduces wrinkles’ appearance without modifying skin parameters such as TEWL, pH, sebum, and melanin content. Volunteers who took part in the study appreciated the CSE-loaded emulsion so much they continued using it in their skincare routine.

## Figures and Tables

**Figure 1 pharmaceutics-13-01634-f001:**
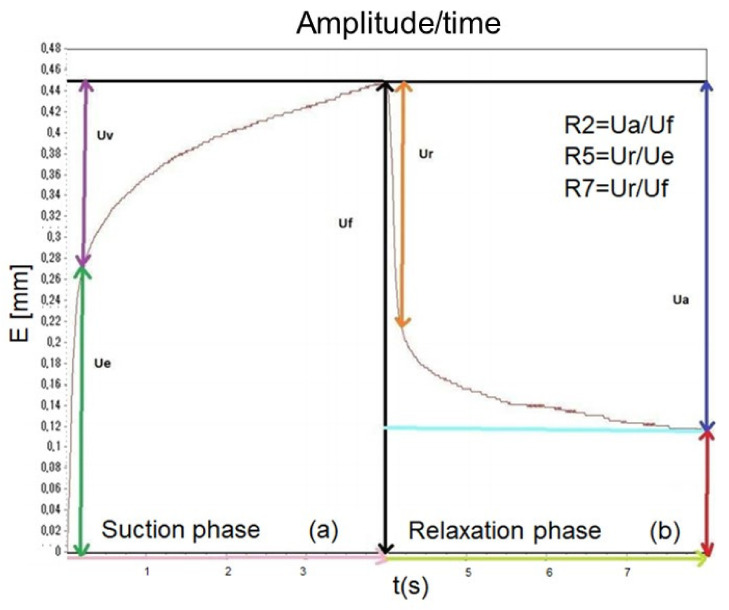
Strain–time curve (deformation, mm/time, s) obtained by Cutometer Dual MPA 580 used to extrapolated elasticity skin parameters (R2, R5, and R7).

**Figure 2 pharmaceutics-13-01634-f002:**
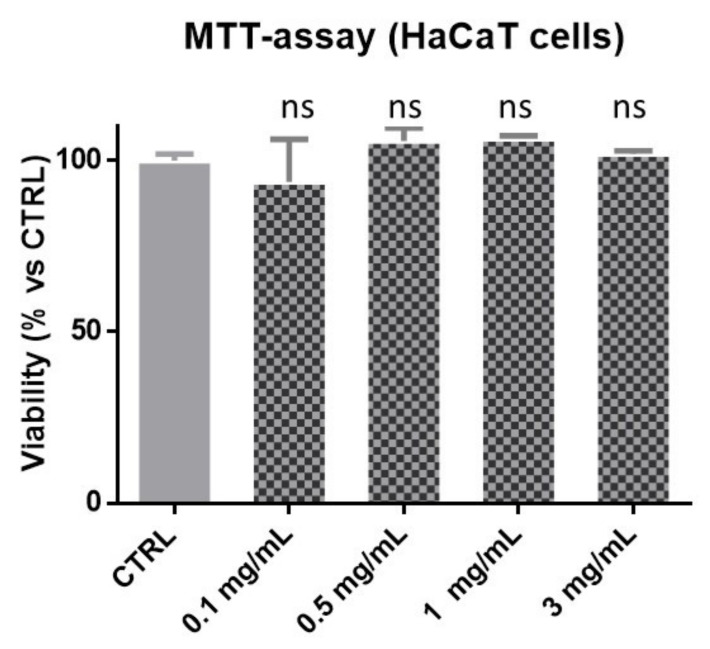
CSE effect on HaCaT cells viability by MTT test. Results shown are the mean ± SD of three determinations. Data were analyzed by one-way ANOVA followed by Tukey’s multiple comparison and data considered significant different *p* value < 0.05. ns = not significant vs. control (CTRL).

**Figure 3 pharmaceutics-13-01634-f003:**
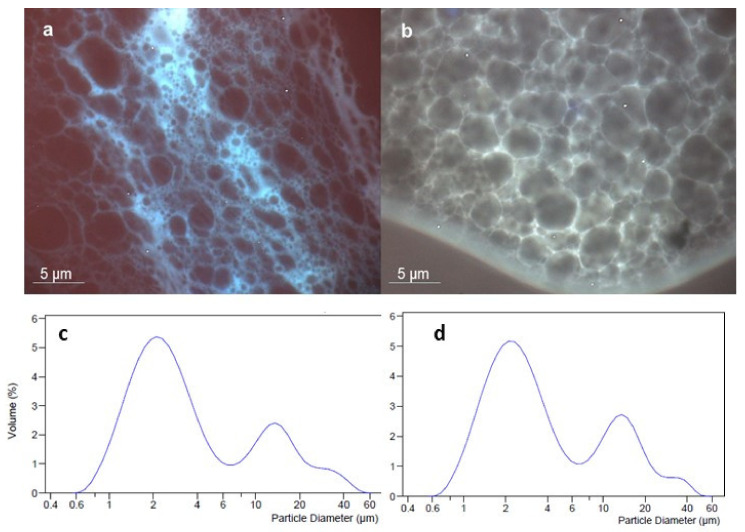
Emulsion system morphology with optical microscopy in fluorescence mode (DAPI filter and darkfield) of blank (**a**) and CSE-loaded (**b**) emulsions and dimensional distribution of oil droplets within the aqueous phase of Blank (**c**) and CSE-loaded (**d**) emulsion.

**Figure 4 pharmaceutics-13-01634-f004:**
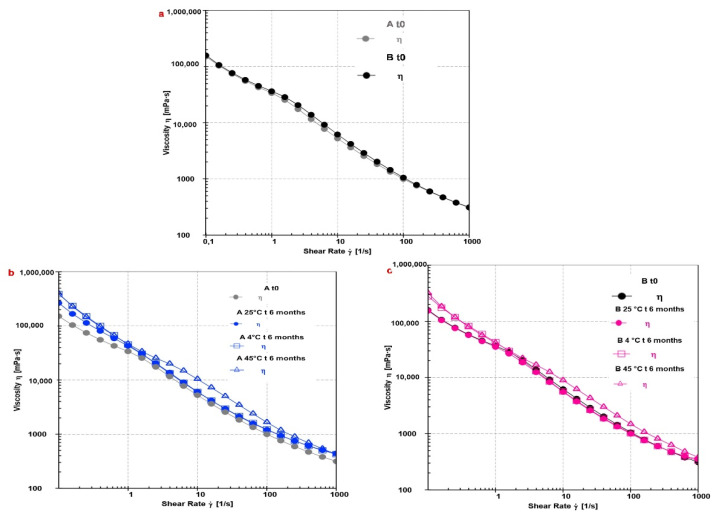
Flow curves obtained from viscosity (η) as a function of shear rate (Ύ), for the different formulations (A and B, (**a**)), and at different storage conditions (A, (**b**), and B, (**c**)).

**Figure 5 pharmaceutics-13-01634-f005:**
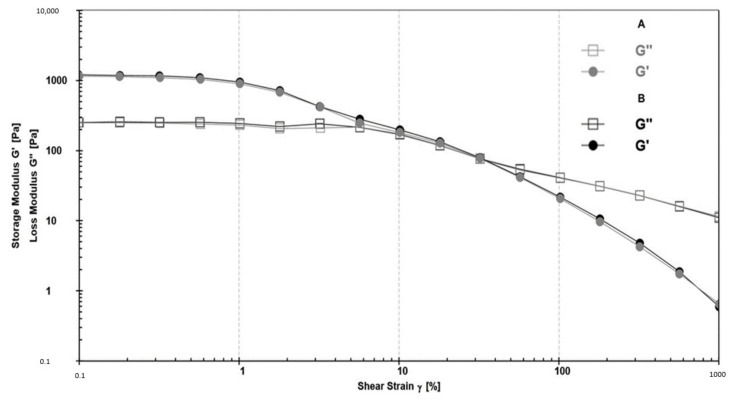
Viscoelastic characterization of A and B represented plotting the storage modulus G′ and loss modulus G″ as a function against complex strain, at constant angular frequency.

**Figure 6 pharmaceutics-13-01634-f006:**
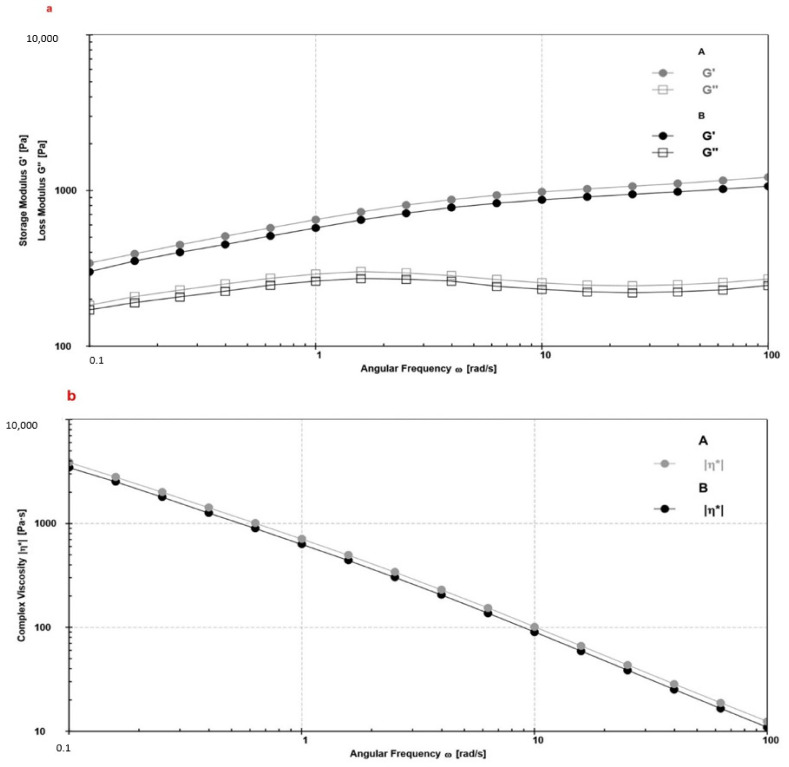
Mechanical spectra of the formulations A and B represented plotting the storage modulus (G′) and loss modulus (G″) (**a**) and complex viscosity (η*) (**b**) as a function of angular frequency, at constant complex strain (0.5%).

**Figure 7 pharmaceutics-13-01634-f007:**
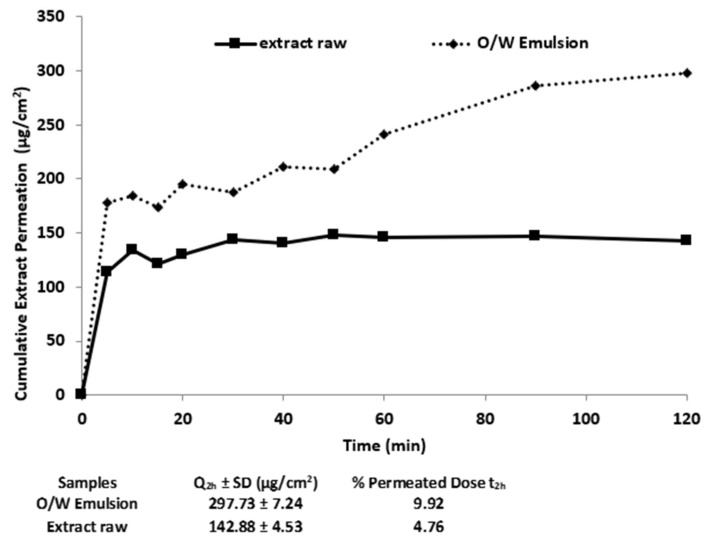
In vitro permeation profiles of CSE from the CSE-loaded emulsion formulation through a synthetic membrane similar to human skin. The control sample was a 50% ethanol solution of raw extract; Q_2h_: extract amount permeation at the end of the test; (mean ± SD, *n* = 3, standard deviation <1% for each point).

**Figure 8 pharmaceutics-13-01634-f008:**
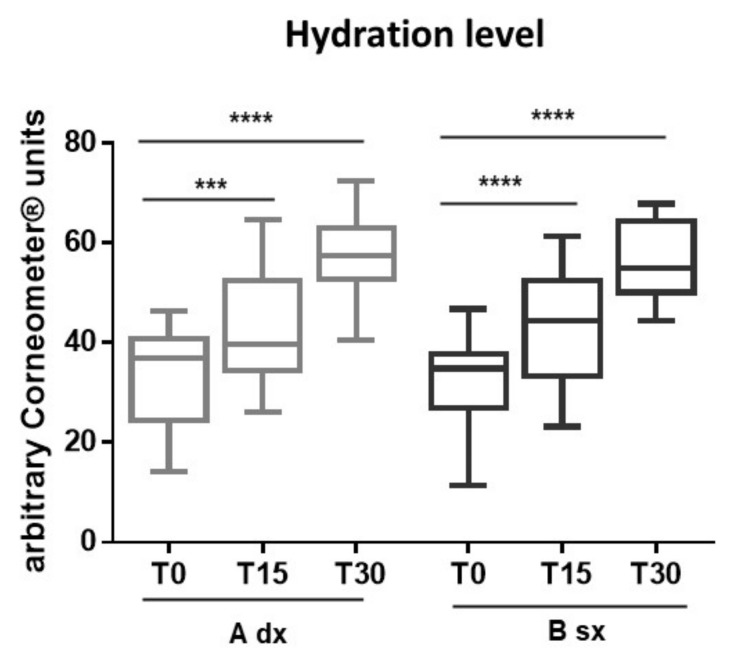
Variation of hydration values before treatments (T_0_) and after 30 days (T_30_) of application of A and B. Results as mean ± SD of three determinations for each volunteer at different measurement times. Data were analyzed by two-way RM (repeated measures) ANOVA (analysis of variance), followed by a Bonferroni adjustment. Data were significant different *p* value < 0.05 (*p* < 0.001 ***; *p* < 0.0001 ****).

**Figure 9 pharmaceutics-13-01634-f009:**
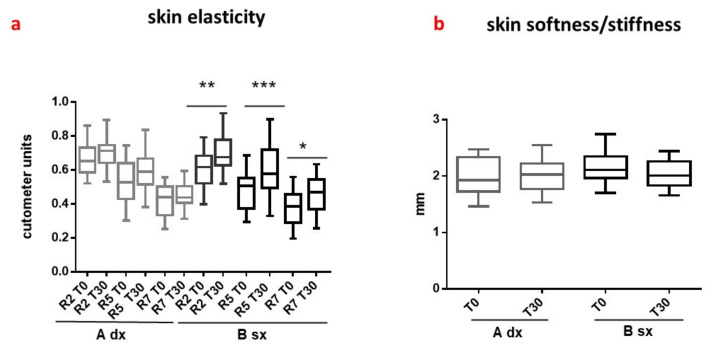
Skin elasticity data (**a**) and skin firmness values (**b**) recorded at time 0 (T_0_, before the application of A and B) and after 30 days (T_30_) of treatment. Results shown are mean ± SD of three determinations for each volunteer at different measurement times. Data were analyzed by two-way RM (repeated measures) ANOVA (analysis of variance), followed by a Bonferroni adjustment. Data were significant different *p* value < 0.05 (*p* < 0.05 *; *p* < 0.01 **; *p* < 0.001 ***).

**Figure 10 pharmaceutics-13-01634-f010:**
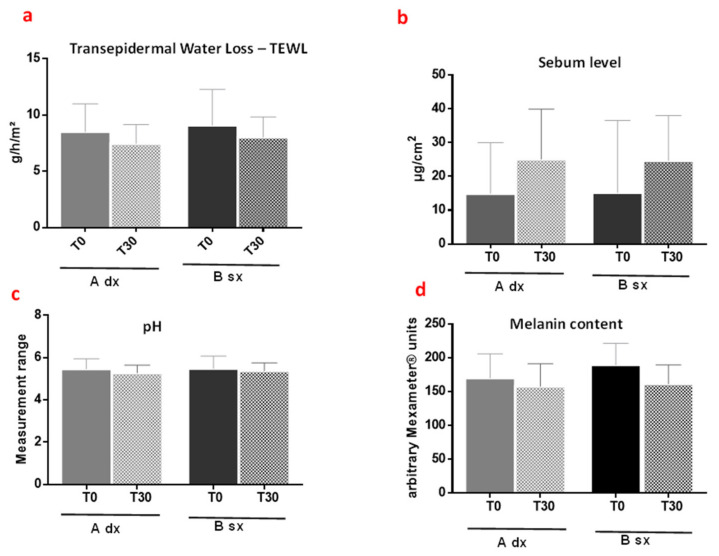
Changes in skin parameters measured ((**a**), TEWL; (**b**), Sebum; (**c**), pH; (**d**), melanin content) after thirty days of application of A and B. Results shown are mean ± SD of three determinations for each volunteer at different measurement times. Data were analyzed by two-way RM (repeated measures) ANOVA (analysis of variance), followed by a Bonferroni adjustment. Data were significant different *p* value < 0.05.

**Figure 11 pharmaceutics-13-01634-f011:**
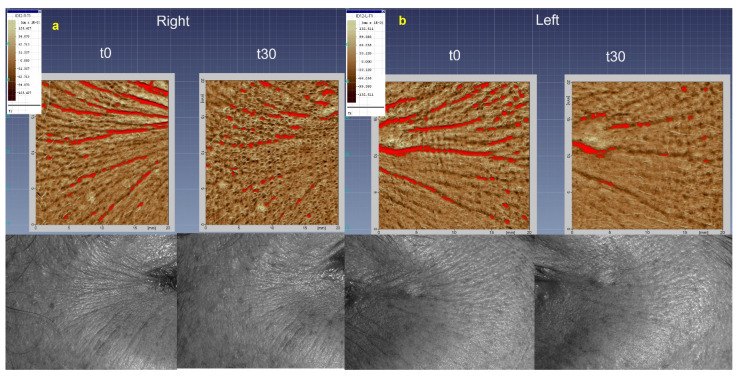
3D DermaTop images (bottom part) and corresponding skin topography (upper part) at t_0_ and t_30_ days. (**a**) corresponds to the right side of the face where the formulation A was applied, and (**b**) corresponds to the left side of the face where the formulation B was applied. The reddish-colored regions indicate the depth of analyzed wrinkles.

**Figure 12 pharmaceutics-13-01634-f012:**
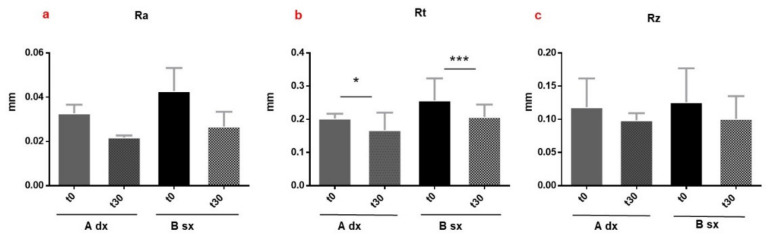
Average of roughness parameters ((**a**), Ra, the mean roughness of the skin; (**b**), Rt, the maximum relief height, and (**c**), Rz, average maximum profile height). Results shown are mean ± SD of three determinations for each volunteer at different measurement times. Data were analyzed by two-way RM (repeated measures) ANOVA (analysis of variance), followed by a Bonferroni adjustment. Data were significant different *p* value < 0.05 (*p* < 0.05 *; *p* < 0.001 ***).

**Figure 13 pharmaceutics-13-01634-f013:**
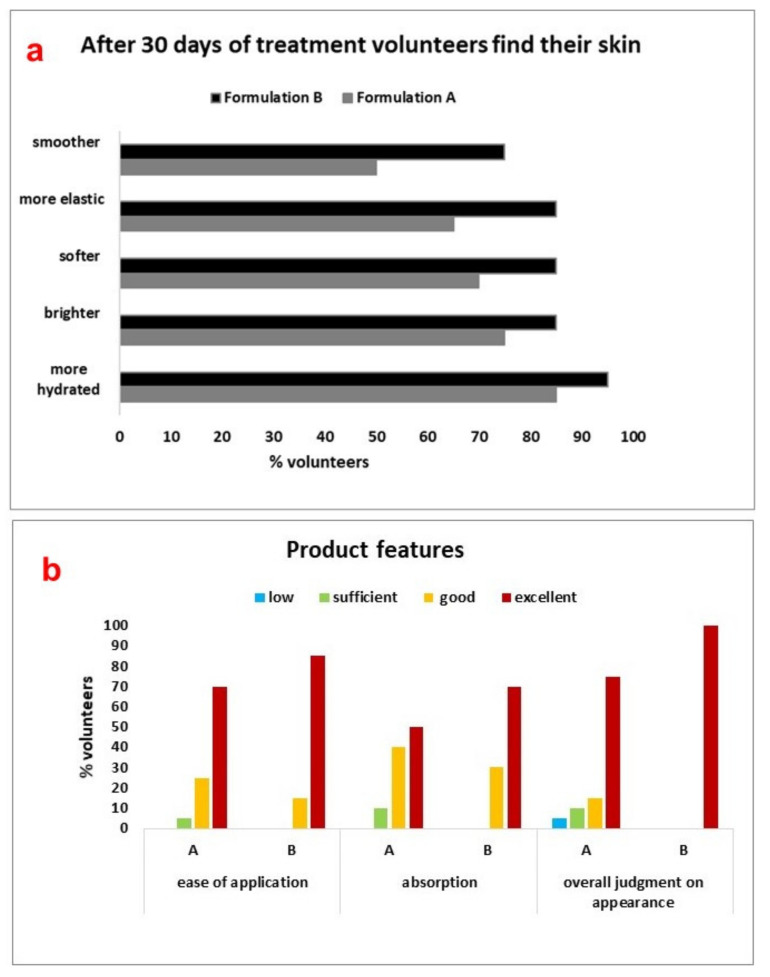
Results obtained from the sensory analysis performed by the volunteers through the self-assessment test, after the application of A and B. (**a**) shows the perception of the skin (elasticity, smoothness, brightness) after the application; (**b**) shows overall opinion (easy application, absorption, and appearance) of the volunteers on the two emulsions.

**Table 1 pharmaceutics-13-01634-t001:** Qualitative–quantitative composition of blank (A) and CSE-loaded (B) O/W emulsion.

Component	Function	Concentration (%, *w*/*w*)	
		A	B
**Oil phase (Phase A)**			
Glyceryl Stearate, PEG-100 Stearate ^1^	Emulsifying	5.0	5.0
Olea Europaea Oil	Emollient	3.0	3.0
Olea Europea Oil Unsaponifiables	Emollient	0.5	0.5
Simmondsia Chinensis Oil	Emollient	1.0	1.0
C_12_-C_15_ Alkyl Benzoate	Emollient	5.0	5.0
Caprylic-Capric Triglycerides	Emollient	4.0	4.0
Decyl Oleate	Emollient	2.5	2.5
Cetearyl Alcohol	Viscosity controlling	2.0	2.0
Dimethicone	Anti-foaming agent	1.0	1.0
**Aqueous phase (Phase B)**			
Purified Water	Diluent	*q.s.* 100	*q.s.* 100
Disodium EDTA	Metal ion binder	0.1	0.1
Glycerin	Humectant	3.0	3.0
Xanthan Gum	Rheology modifier	0.1	0.1
Carbomer	Rheology modifier	0.2	0.2
**Phase C**			
CSE	New Active ingredient	-	0.3
**Phase D**			
Phenoxyethanol, Benzoic Acid, Dehydroacetic Acid, Ethylhexylglycerin ^2^	Preservative	0.6	0.6
Imidazolidinyl Urea	Preservative	0.3	0.3
**Phase E**			
Aluminium Starch Octenylsuccinate	Absorbent agent	1.0	1.0
Glycerin	Humectant	1.0	1.0
**Phase F**			
l-arginine	Buffering	*q.s.* to pH = 5.5	*q.s.* to pH = 5.5

*q.s.* = quantum sufficit; ^1^ The breakdown of this ingredient is Glyceryl Stearate (50%), PEG-100 Stearate (50%); ^2^ Phenoxyethanol, Benzoic Acid, Dehydroacetic Acid, Ethylhexylglycerin are the ingredients of Euxyl^®^ K 701, a liquid cosmetic preservative. The order is based on the concentration used for the mixture.

**Table 2 pharmaceutics-13-01634-t002:** Product classification based on the medium irritation index.

Average Irritation Index (MII)	Class
From 0.0 to 0.5	Not irritating
from 0.5 to 2.0	Slightly irritating
from 2.0 to 5.0	Moderately irritating
from 5.0 to 8.0	Strongly irritating

**Table 3 pharmaceutics-13-01634-t003:** pH values of formulations A (blank) and B (CSE-loaded) from 24 h to 6 months after preparation at different storage temperatures.

Time (Days)	Formulation					
	A			B		
	4 °C	25 °C	45 °C	4 °C	25 °C	45 °C
1	5.57 ± 0.01	5.56 ± 0.02	5.57± 0.01	5.44 ± 0.02	5.44 ± 0.02	5.44 ± 0.01
15	5.58 ± 0.02	5.58 ± 0.01	5.57 ± 0.02	5.51 ± 0.01	5.58 ± 0.02	5.54 ± 0.02
30	5.48 ± 0.02	5.49 ± 0.01	5.49 ± 0.01	5.47 ± 0.01	5.43 ± 0.01	5.50 ± 0.01
60	5.57± 0.01	5.59 ± 0.02	5.58 ± 0.01	5.53 ± 0.02	5.55 ± 0.01	5.53 ± 0.02
180	5.47 ± 0.03	5.48 ± 0.03	5.49 ± 0.02	5.45 ± 0.01	5.43 ± 0.03	5.46 ± 0.01

No statistical difference (*p* > 0.05).

## Data Availability

Not applicable.

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
