# Peer review of "Development, Characterization, and Clinical Investigation of a New Topical Emulsion System Containing a Castanea sativa Spiny Burs Active Extract"

_pharmaceutics, 2021, doi:10.3390/pharmaceutics13101634_

Round 1

Reviewer 1 Report

The paper describes a skin care cream containing a natural extract, its characterization and testing on human subjects. This is a very complete and detailed study.

The title should be revised-delete newly or change to new

The abstract could do more to highlight novelty.  As it is a new API, the desired activity and target could be clearly stated. It states that there is a skin delivery system, but not its purpose nor active concentration. 

Line 97. Consider rewording this statement and explaining its relevance to the current work. 

Line 105. Instead (of what)? It Is hard to follow rational

As this is a natural extracted product, there should be characterization details included regarding total; phenolic and flavonoid content (or refer to previous reported characterisation)

Lone 301. Add references to support use of q quercetin-3-O-β-D-glucoside, 300 used as a marker of CSE

327, performances

335 “The inclusion criteria were verified by the dermatologist after observation of the volunteer and registration of the information provided by himself.” It may be useful to introduce the role of the dermatologist and explain this statement

215 define MTT test

730 include limits for classification as non-irritating

829 wrinkle depth?

Figure 13. Modify title to remove reference to gender

Line 863 smooth

There are a number of typographical errors throughout and it does require detailed proof-reading.

Author Response

The authors thank for the chance to increase the quality of the work.  The manuscript has been revised following your suggestions as follows:

Reviewer 1

Comments and Suggestions for Authors

The paper describes a skin care cream containing a natural extract, its characterization and testing on human subjects. This is a very complete and detailed study.

The title should be revised-delete newly or change to new

As you suggested the title has been revised.

The abstract could do more to highlight novelty.  As it is a new API, the desired activity and target could be clearly stated. It states that there is a skin delivery system, but not its purpose nor active concentration. 

As you suggested the abstract has been totally re-written

Line 97. Consider rewording this statement and explaining its relevance to the current work. 

As you suggested the sentence has been rewritten. 

Line 105. Instead (of what)? It Is hard to follow rational

The word “Instead” has been removed.

As this is a natural extracted product, there should be characterization details included regarding total; phenolic and flavonoid content (or refer to previous reported characterisation)

the authors apologize. In the revised text the reference to the extract chemical profile reported in their previous research (Esposito et al. 2019, reference n. 6) has been added

Lone 301. Add references to support use of q quercetin-3-O-β-D-glucoside, 300 used as a marker of CSE

The reference (Esposito et al. 2019, reference n. 6) has been added

327, performances

The mistake has been corrected

335 “The inclusion criteria were verified by the dermatologist after observation of the volunteer and registration of the information provided by himself.” It may be useful to introduce the role of the dermatologist and explain this statement

The role of dermatologist has been described and the statement has been explained.

215 define MTT test

The definition has been added

730 include limits for classification as non-irritating

As you required, the limits have been added

829 wrinkle depth?

Sorry, the mistake has been corrected

Figure 13. Modify title to remove reference to gender

The figure title has been modified

Line 863 smooth

The mistake has been corrected

There are a number of typographical errors throughout and it does require detailed proof-reading.

Sorry, now a detailed proof -reading has been performed.

Reviewer 2 Report

Dear Authors,

The manuscript entitled "Development, characterization and clinical investigation of a newly topical emulsion system containing a Castanea sativa spiny burs active extract" reports an interesting work about the development of an O/W emulsion for skincare containing Castanea sativa as an antioxidant agent. The emulsion was evaluated on healthy human volunteers. The authors concluded that the emulsion was stable and had suitable physicochemical properties for dermal application and they observed an improvement in the skin biomechanical properties such as hydration, skin elasticity and a reduction in the periorbital wrinkles in 30 days. 

The manuscript is well written and well-organized. The introduction provides enough background, and relevant references have been cited. The work has been well-planned and the methodology is properly described. The results are clearly presented in general. However, I have a few comments that, in my opinion, may contribute to enhance the quality of the manuscript:

  1. It is a good and extensive work, however, in my opinion, the abstract does not reflect the importance of the work. I think is important to put the reader in the context of what the formulation developed, characterized and evaluated is intended for. For instance, in line 20 something like "for skincare" could be added after ingredient-based products. 
    "Technological innovation and rational use of natural resources can reduce wastes, provide 18 social and environmental benefits, and act as a catalyst to attentive consumers in selecting natural 19 ingredients-based products for skincare."
  2. Table 1 and 2: please revise their format and adapt it to the Journal's format. 
  3. Figure 2: According to section 2.3, lines 181-182, the data are reported in %, the title of Y-axis in Figure 2 also indicates viability in %, however, the scale is expressed as a decimal. please review. 
  4. Lines 271, 273, 282 and 303 please indicate the number of the equation (1), (2)... and so on. 
  5. Line 303, please describe below the equation what the abbreviations stand for. 
  6. Figures 5 and 6: It is unclear what line corresponds to G' and which one to G", please use different symbols. 
  7. Figure 7: please review the title of the Y-axis. ug/cm2 are commonly units of permeated drug amount; and the flux ug/min.cm2; please review and correct the title if applies. Additionally, it is not cited in text, please include it. 
  8. Figure 8: the legend in figure 8 states that the results are expressed as the "mean +/- SD of three determinations". Please clarify if they were 3 determination for each volunteer.  
  9. Line 805: please review the range reported in lines 805-806. According to Figure 10b, the range is about 15-25 ug/cm2, and the range in line 806 (10c) is missing. 
  10. Line 814 in the legend of figure 8: The sentence "Results as mean ± SD of three determinations" is duplicated, please review.
  11. Lines 832-834: please review the values reported. based on Figure 12, it seems the data have been swapped between panels b and c. 
  12. Figure 11: in my opinion, adding the description in the figure legend that panel a corresponds to the right side of the face where the formulation A was applied, and panel b corresponds to the left side of the face where the formulation B was applied will ease the reader to interpret and visualize the results. 

Kind regards,

Author Response

The authors thank for the chance to increase the quality of the work. The manuscript has been
revised following your suggestions as follows:
Reviewer 3:
Dear Authors,
The manuscript entitled "Development, characterization and clinical investigation of a newly topical
emulsion system containing a Castanea sativa spiny burs active extract" reports an interesting work
about the development of an O/W emulsion for skincare containing Castanea sativa as an
antioxidant agent. The emulsion was evaluated on healthy human volunteers. The authors
concluded that the emulsion was stable and had suitable physicochemical properties for dermal
application and they observed an improvement in the skin biomechanical properties such as
hydration, skin elasticity and a reduction in the periorbital wrinkles in 30 days.
The manuscript is well written and well-organized. The introduction provides enough background,
and relevant references have been cited. The work has been well-planned and the methodology is
properly described. The results are clearly presented in general. However, I have a few comments
that, in my opinion, may contribute to enhance the quality of the manuscript:
1. It is a good and extensive work, however, in my opinion, the abstract does not reflect the
importance of the work. I think is important to put the reader in the context of what the
formulation developed, characterized and evaluated is intended for. For instance, in line 20
something like "for skincare" could be added after ingredient-based products.
"Technological innovation and rational use of natural resources can reduce wastes, provide
18 social and environmental benefits, and act as a catalyst to attentive consumers in
selecting natural 19 ingredients-based products for skincare."
As you suggested the abstract and its content has been totally revised
2. Table 1 and 2: please revise their format and adapt it to the Journal's format.
The format of Table 1 and 2 has been adapted
3. Figure 2: According to section 2.3, lines 181-182, the data are reported in %, the title of Yaxis in Figure 2 also indicates viability in %, however, the scale is expressed as a decimal.
please review.
Figure 2 has been revised according to your suggestion
4. Lines 271, 273, 282 and 303 please indicate the number of the equation (1), (2)... and so
on.
The numbers (1-4) of equations have been added
5. Line 303, please describe below the equation what the abbreviations stand for.
We apologize for the lack of information. The abbreviations have been detailed in the revised text
2
6. Figures 5 and 6: It is unclear what line corresponds to G' and which one to G", please use
different symbols.
Sorry, in the revised Figures 5 and 6 different symbols have been used
7. Figure 7: please review the title of the Y-axis. ug/cm2 are commonly units of permeated
drug amount; and the flux ug/min.cm2; please review and correct the title if applies.
Additionally, it is not cited in text, please include it.
Thank you for the advice, the Y-axis has been revised and the figure has been cited in the text
8. Figure 8: the legend in figure 8 states that the results are expressed as the "mean +/- SD of
three determinations". Please clarify if they were 3 determination for each volunteer.
The clarification has been added in the legend of figure 8
9. Line 805: please review the range reported in lines 805-806. According to Figure 10b, the
range is about 15-25 ug/cm2, and the range in line 806 (10c) is missing.
Sorry, the mistakes have been corrected
10. Line 814 in the legend of figure 8: The sentence "Results as mean ± SD of three
determinations" is duplicated, please review.
The duplication has been removed
11. Lines 832-834: please review the values reported. based on Figure 12, it seems the data
have been swapped between panels b and c.
The data have been checked
12. Figure 11: in my opinion, adding the description in the figure legend that panel a
corresponds to the right side of the face where the formulation A was applied, and panel b
corresponds to the left side of the face where the formulation B was applied will ease the
reader to interpret and visualize the results.
Thanks, the description has been added.
Kind regards,

Reviewer 3 Report

Development, characterization, and clinical investigation of a newly topical emulsion system containing a Castanea sativa 3 spiny burs active extract. A major revision should be done before publication.

The abstract should be rewritten to show an appropriate summary of the article. There is irrelevant information in it.

Section 2.1 please provide the company details of the raw materials.

Table 1. authors claimed that the antioxidant properties of the emulsion come from the vegetable extract, however in table 1 appear several compounds that could have an antioxidant property as well, i.e. olive oil. Olive oil unsaponificables, simmondsia chinensis oil, decil oleate. In addition, the vegetable extract concentration is relatively low (0.3%), then infer that the effect is due to the vegetable extract considering the possible lack of statistical differences between emulsions (which was no carried out appropriately, as followed discussed)

Why did authors add two different preservatives? They carried out a challenge test to demonstrated that it is necessary?

Section 2.5. Authors added DAPI in the microscopy study? Please included the concentration/dilution used.

Line 224, what is the relative humidity of climatic condition 25ºC? why authors choose 45ºC for the accelerated condition? The international guidelines ICH (stability studies) specify 40ºC. please indicate the reason of this discrepancy.

Section 2.7. Why authors quantified quercetin, which is a flavanol if they look for the penetration of a polyphenol? They should use a polyphenol, based on the article claims. Please refer to the LOQ of the analytical method. In addition, a clarification of the terminology vegetable compounds polyphenol, flavonoid, tannins, etc should be given to improve understanding and uniform terminology. Line 507-508 how authors know that the two main components are tannins, quercetin, flavonoids if the polyphenol content is expressed in gallic acid. These aspects are a little bit confusing.

It is not clear to me what the authors want to claim in the clinical studies, in the introduction many effects are described, but in line 734 seem to be focus on photoaging. What is the relationship with the test carried out and the active ingredient (polyphenol?) polyphenol could regulate sebum, melanin, wrinkles, elasticity, firmness, hydration, everything at the same time? What is the relationship of the polyphenol with the TEWL and with the pH? I think that the studies should be more focus based on the product claim. If authors want to measure the effect on photoaging, why they tested pH, sebum, TEWL, etc. Reading the discussion, it is a little bit more clear that some parameters are related with efficacy and others with safety. Please clarify from the beginning.

Section 2.10. the experimental design as far as understood is an intra-patient control (one side of the body is treated with the active and the other one with the placebo) at different time-points, then the statistical test should be based on this design, a repetitive measure ANOVA considering intra and inter-patient analysis (or mixed effect model) should be carried out. In addition, I can see a minimum of two independent factors, measurement time and formulation.

Figure 2. statistical analysis should be added.

Section 3.3. the droplet size of the figure 3a seems to be a little bit not homogeneous. Please provide the mean droplet size and polydispersion. Line 560 (also line 704), micelles are usually in the nanometric range, probably what author observe are lipid droplet and not micelles.

Line 586. The crystallization of the water phase would probably occur under 0ºC. please clarify.

Line 589. Could authors state the extrapolation prediction of the physical stability considering the selected centrifugation parameters selected?

Line 614. Please provide the reference where appeared the link between rheological parameters and the cosmetic attributes.

Figure 4. please improve legend readability.

Line 644. Please include the reference in an appropriate way, according to the other references.

Figure 6. please make legend more readable. In addition, figure 6A have the same colour for A and B and it is difficult to distinguish. 

Figure 6 A did not show a crossover the point, but figure 5 did. Please discuss accordingly.

Line 716-717 please explain the rational about the extrapolation of efficacy concentrations (ug/mL) and the amount permeated (ug/cm2). Also please include the percentage of the dose applied at the end of the experiment. Also, the representation in figure 7 is more similar to a release profile than a permeation profile, also considering that an infinite dose was applied to the donor compartment. Could authors explain the reason of the profile if the membrane resembles the skin behaviour?

Line 754-756. I disagree with author statement. First of all if they want to claim differences between formulation, a proper statistical analysis should be carried out, as I commented previously. According to box plot in figure 8, the mean value of T30 for creams A and B are similar, the discussion should be corrected. Authors should not magnify their results only for the maximum value, all data and the statistical analysis must be considered. According to the plot I have doubts about the superiority of cream B over A, and this could be explained by my previous comment about the composition of creams A and B. The same comments apply for al clinical studies carried out. A proper statistical method must be use and discussion must be based on the statistical results.

Author Response

The authors thank for the chance to increase the quality of the work. The manuscript has been revised following your suggestions as follows:

Reviewer 2

Comments and Suggestions for Authors

Development, characterization, and clinical investigation of a newly topical emulsion system containing a Castanea sativa spiny burs active extract. A major revision should be done before publication.

The abstract should be rewritten to show an appropriate summary of the article. There is irrelevant information in it.

The abstract has been totally rewritten

Section 2.1 please provide the company details of the raw materials.

The company details have been added.

Table 1. authors claimed that the antioxidant properties of the emulsion come from the vegetable extract, however in table 1 appear several compounds that could have an antioxidant property as well, i.e. olive oil. Olive oil unsaponificables, simmondsia chinensis oil, decil oleate. In addition, the vegetable extract concentration is relatively low (0.3%), then infer that the effect is due to the vegetable extract considering the possible lack of statistical differences between emulsions (which was no carried out appropriately, as followed discussed)

The authors apologize if from the discussion of the results it was deduced that the only functional ingredient of the formulation was the plant extract. In the revised text we have explicitly indicated that there is a synergy of action between the new extract and the selected ingredients (paragraph "O / W Emulsion Formulation). Moreover, in the paragraph “CSE and Emulsions physico-chemical stability test” we have indicated that the evaluation of the chemical stability of polyphenols is correlated. to the antioxidant activity of the plant extract and not of the whole formulation. Furthermore, according to your indications the statistical analysis of the clinical tests has been revised.

Why did authors add two different preservatives? They carried out a challenge test to demonstrated that it is necessary?

The authors did not conduct the Challenge test. The combination of the two preservatives was chosen to ensure a broad spectrum of activity. In particular, the Imidazolidinyl Urea is more active on bacteria Gram +, Gram - and Yeast. Euxyl is also active against Mold fungi (Aspergillus niger and Penicillium funiculosum)

Section 2.5. Authors added DAPI in the microscopy study? Please included the concentration/dilution used.

The optical fluorescence microscope used is directly equipped with a set of three filters housed in the fluorescence “filter cube”. As excitation filter was selected DAPI Ultraviolet excitation in wavelength range 450-500, blue – aqua color. This explanation has been added to the revised text.

Line 224, what is the relative humidity of climatic condition 25ºC? why authors choose 45ºC for the accelerated condition? The international guidelines ICH (stability studies) specify 40ºC. please indicate the reason of this discrepancy.

Thanks for your considerations. The relative humidity has been added. The authors carried out the accelerated stability tests according to Guidelines on stability testing of cosmetic products (by Cosmetics Europe, the european trade association representing the interest of the cosmetics, toiletry and perfumery industry 2004, reference n 27) in which it is indicated that the stability of a cosmetic product can be predicted using different conditions and temperatures, and the "Tests are often performed at 37 ° C, 40 ° C or 45 ° C during 1, 2, 3 ... months ". The missing reference has been added in the revised manuscript

Section 2.7. Why authors quantified quercetin, which is a flavanol if they look for the penetration of a polyphenol? They should use a polyphenol, based on the article claims. Please refer to the LOQ of the analytical method.

Flavonols, such as quercetin, are polyphenols belonging to the flavonoid family (Soto et al. 2015, reference n 42). The LOQ value has been added

In addition, a clarification of the terminology vegetable compounds polyphenol, flavonoid, tannins, etc should be given to improve understanding and uniform terminology. Line 507-508 how authors know that the two main components are tannins, quercetin, flavonoids if the polyphenol content is expressed in gallic acid. These aspects are a little bit confusing.

The authors apologize for the confusion. In the revised text the terms concerning plant compounds have been clarified. Moreover, the authors know the chemical profile of the extract because they investigated it in their previous research (Esposito et al 2019, reference n 6). In the revised text the bibliography is reported.

It is not clear to me what the authors want to claim in the clinical studies, in the introduction many effects are described, but in line 734 seem to be focus on photoaging.

Thanks for your remark. In the revised text already in the introduction, the authors refer to photoaging, because low hydration, elasticity, firmness of the skin and the appearance of wrinkles are all signs of premature aging (photoaging).

What is the relationship with the test carried out and the active ingredient (polyphenol?) polyphenol could regulate sebum, melanin, wrinkles, elasticity, firmness, hydration, everything at the same time? What is the relationship of the polyphenol with the TEWL and with the pH? I think that the studies should be more focus based on the product claim. If authors want to measure the effect on photoaging, why they tested pH, sebum, TEWL, etc. Reading the discussion, it is a little bit more clear that some parameters are related with efficacy and others with safety. Please clarify from the beginning.

We have verified many potential antiaging effects of our formulations because the increase in melanin content, the appearance of wrinkles, the reduction of elasticity, firmness, and hydration, are skin alterations due to oxidative stress and the overproduction of free radicals (Soto et al. 2015, Jadoon 2015, references n 42 and 4). Natural polyphenols possess scavenging properties towards radical oxygen species that make them interesting for antiaging purposes in cosmetics. In the Discussion section, the authors have clarified the parameters related to formulations efficacy and the parameters evaluated to demonstrate the formulations safety.

Section 2.10. the experimental design as far as understood is an intra-patient control (one side of the body is treated with the active and the other one with the placebo) at different time-points, then the statistical test should be based on this design, a repetitive measure ANOVA considering intra and inter-patient analysis (or mixed effect model) should be carried out. In addition, I can see a minimum of two independent factors, measurement time and formulation.

Thanks for your considerations. As you suggested in the revised manuscript the results of in vivo clinical tests have been analysed by analyzed by a two-way RM (repeated measures) ANOVA (analysis of variance), followed by a Bonferroni adjustment to determine the statistically significant differences between the measurement time and formulation (p ≤ 0.05). Based on your suggested statistical analysis, figures 8, 9 and 12 have been replaced. On skin elasticity and on the wrinkle depth index, a better result seems to remain with the application of cream B. According to the new darta the text has been sligtly revised.

Figure 2. statistical analysis should be added.

The statistical analysis has been added as ns=not significant Vs control (CTRL)

Section 3.3. the droplet size of the figure 3a seems to be a little bit not homogeneous. Please provide the mean droplet size and polydispersion.

As you suggest the dimensional distribution analysis and SPAN determination have been added and the discussion has been revised at the section 3.3 and revised figure 3. The paragraph describing the LLS method has been added to the section 2.5.

Line 560 (also line 704), micelles are usually in the nanometric range, probably what author observe are lipid droplet and not micelles.

Thank you for the observation. We mistakenly used the word “micelles”, now replaced with “oil droplets” along the revised text.

Line 586. The crystallization of the water phase would probably occur under 0ºC. please clarify.

Sorry, the sentence has been re-written

Line 589. Could authors state the extrapolation prediction of the physical stability considering the selected centrifugation parameters selected?

The Cosmetics Europe Guidelines on Stability Testing of Cosmetic Products 2004 state the need to subject the cosmetic product to mechanical stress without however indicating the experimental conditions. In our case, the conditions of the centrifuge test were chosen on the basis of our experimental know-how (reference n 25). The used conditions (two successive cycles on the same emulsion sample, the first at 4000 rpm for 30 minutes and the second at 5300 rpm for 15 minutes) seem to be more drastics than those reported in other researches in which the stability of emulsions loaded with plant extracts has been evaluated. For example, Sememzato et al. 2018 (reference n 12) used 1 cycle at 4800 rpm for 30 minutes; Daudt et al. 2015 (reference n 43) used 1 cycle at 3000 rpm for 30 minutes.

Line 614. Please provide the reference where appeared the link between rheological parameters and the cosmetic attributes.

As you suggested, the reference has been added (reference n 46)

Figure 4. please improve legend readability.

The legend has been revised

Line 644. Please include the reference in an appropriate way, according to the other references.

Sorry, the reference has been revised accordin to your suggestion

Figure 6. please make legend more readable. In addition, figure 6A have the same colour for A and B and it is difficult to distinguish.

Thanks, figure 6 has been revised

Figure 6 A did not show a crossover the point, but figure 5 did. Please discuss accordingly.

In the revised text a brief discussion has been added.

Line 716-717 please explain the rational about the extrapolation of efficacy concentrations (ug/mL) and the amount permeated (ug/cm2).

We apologize for the lack of information. The values in the permeation equation that translate the concentration from µg /µl to µg /cm2 have been added to section 2.7 in the revised text. The amount permeated in the unit time per area is higher than the active free radical scavenging dose, so it could explain the efficacy.

Also please include the percentage of the dose applied

As reported in material and methods section the applied dose was of 1 g of formulation with an extract content of 3 mg. In percentage was of 0.3%. As you suggested, this information has been added to the revised text

Also, the representation in figure 7 is more similar to a release profile than a permeation profile, also considering that an infinite dose was applied to the donor compartment. Could authors explain the reason of the profile if the membrane resembles the skin behaviour?

Authors do not understand what the revisor means saying “the representation in figure 7 is more similar to a release profile than a permeation profile”. For the test we used a defined amount (1g) of formulation on donor compartment and the obtained profile is related to the amount of extract permeated through the membrane and recovered into acceptor compartment. It was expressed as the amount permeated per surface area of Strat-M, and not as total amount of extract released by the emulsion in physiological conditions. The discussion of permeation results has been slightly revised to better clarify this statement.

Line 754-756. I disagree with author statement. First of all if they want to claim differences between formulation, a proper statistical analysis should be carried out, as I commented previously. According to box plot in figure 8, the mean value of T30 for creams A and B are similar, the discussion should be corrected. Authors should not magnify their results only for the maximum value, all data and the statistical analysis must be considered. According to the plot I have doubts about the superiority of cream B over A, and this could be explained by my previous comment about the composition of creams A and B. The same comments apply for al clinical studies carried out. A proper statistical method must be use and discussion must be based on the statistical results.

As you suggested a proper statistical analysis has been carried out. The discussion related to Figure 8 and other clinical results has been corrected, considering all data and the statistical analysis. According to the suggested statistical analysis the differences between A and B have been reconsidered.

Round 2

Author Response

Why did authors add two different preservatives? They carried out a challenge test to demonstrated that it is necessary?

The authors did not conduct the Challenge test. The combination of the two preservatives was chosen to ensure a broad spectrum of activity. In particular, the Imidazolidinyl Urea is more active on bacteria Gram +, Gram - and Yeast. Euxyl is also active against Mold fungi (Aspergillus niger and Penicillium funiculosum)

I disagree with the rational followed regarding the preservative use. In general, the preservatives should be the minimum to maintain a microbiology safe product along the time. SCCS penalize the preservative exposure with a global preservative exposure in diary life and a safety assessment (considering as well a long-term exposure), with a special consideration about preservatives should be carried out before lunching the product. The safety of the consumer should be the first thing in the mind of a cosmetic developer. Then the preservative and its level should be the minimum and selected by a challenge test

The authors are sorry that the first answer was incomplete for you.

The chosen preservatives are listed among those allowed in cosmetic products by the European regulation of cosmetic products (REGULATION (EC) No 1223/2009, ANNEX V) and have been used at concentrations lower than the maximum permitted for leave-on products (such as creams, max. 1.25 % and 0.6%, Data sheet of manifacture company and ANNEX V). Anyway, to be sure of the safety of use, before testing the cream on volunteers, the authors calculated the MoS (Margin of Safety) (REGULATION (EC) No 1223/2009, ANNEX I COSMETIC PRODUCT SAFETY REPORT) and the ​​obtained value resulted higher than 100 as the regulation accepted safety limit (>100) after single application or repeated applications. Having verified the conformity of MoS, the experimental design was directed to other aspects, to arrive at clinical tests.

We hope you agree that determine the minimum concentration of efficacy of the preservative system for each emulsion would have required numerous formulation attempts and on each one to perform a Challenge test should changed the purpose of the research.

Also please include the percentage of the dose applied As reported in material and methods section the applied dose was of 1 g of formulation with an extract content of 3 mg. In percentage was of 0.3%.

 As you suggested, this information has been added to the revised text

Point partially solved.

Sorry If I did not explain properly, what I mean is: if the maximum released value for emulsion is around 300 ug/cm2 and the solution is about 150 ug/cm2. This 300 ug/cm what percentage of the applied dose represent? What is the released/permeated percentage considering the applied dose?

The value in percentage of permeated dose at the end of the test for emulsion and the raw extract was of 9.92 and 4.76%, respectively. This information has been added to the revised figure 7 and to the revised text.

Also, the representation in figure 7 is more similar to a release profile than a permeation profile, also considering that an infinite dose was applied to the donor compartment. Could authors explain the reason of the profile if the membrane resembles the skin behaviour?

 Authors do not understand what the revisor means saying “the representation in figure 7 is more similar to a release profile than a permeation profile”. For the test we used a defined amount (1g) of formulation on donor compartment and the obtained profile is related to the amount of extract permeated through the membrane and recovered into acceptor compartment. It was expressed as the amount permeated per surface area of Strat-M, and not as total amount of extract released by the emulsion in physiological conditions. The discussion of permeation results has been slightly revised to better clarify this statement.

Sorry again for not explain properly. The typical permeation profile is like this image

The straight line represents the steady state that is usually achieved when a high dose is in the donor compartment (as in your case, 1g). From this straight line the transdermal flux is calculated. This high dose is the infinite dose scheme, cited in different transdermal studies and in OCDE guidelines. However, your profile is more similar to a release profile (typical release profile in the following image). Is an accumulative profile and there is no state-state.

If the proposed membrane mimics the skin behaviour, why you did not get the typical permeation profile of the first image? One thing is the release profile and other different thing is the permeation profile. The appropriate terminology should be used in the manuscript.

Authors thank the revisor to better explain his/her scientific point of view. Authors agree that OCDE guidelines give an indication on the applicable quantity to define a finite or infinite dose but, as also you refer to other studies, such scientific reports, reviews or studies have shown that defining an appropriate dose for such investigations can be problematic so sometimes, even infinite dosing may be appropriate if large amounts of formulation are to be used (Selzer et al., 2013; Finite and infinite dosing: Difficulties in measurements, evaluations and predictions; Advanced drug delivery reviews). In our case, the amount loaded on donor compartment was due to the impossibility of uniformly covering the membrane with a smaller quantity ensuring the application of a homogeneous distribution. This sentence has been added to the revised text (2.7 section)  

Moreover, authors agree that the representative permeation behaviour in the figure 7 is a cumulative curve (equation number 4 at 2.7 section). The cumulative curve to describe the permeation behavior is assumed by many authors (Montenegro et al., 2017 In Vitro Evaluation of Sunscreen Safety: Effects of the Vehicle and Repeated Applications on Skin Permeation from Topical Formulations – Pharmaceutics; Kim et al 2021 Preparation of a Capsaicin-Loaded Nanoemulsion for Improving Skin Penetration; Journal of agricultural and food chemistry) to describe the permeation profile.

Anyway, as you suggested to better clarify, authors revised the figure 7 inserting “cumulative extract permeation” on the y-axis, also adding to the graph the values of % Dose permeated at the end of the test.

Section 2.10. the experimental design as far as understood is an intra-patient control (one side of the body is treated with the active and the other one with the placebo) at different time-points, then the statistical test should be based on this design, a repetitive measure ANOVA considering intra and inter-patient analysis (or mixed effect model) should be carried out. In addition, I can see a minimum of two independent factors, measurement time and formulation.

Thanks for your considerations. As you suggested in the revised manuscript the results of in vivo clinical tests have been analysed by analyzed by a two-way RM (repeated measures) ANOVA (analysis of variance), followed by a Bonferroni adjustment to determine the statistically significant differences between the measurement time and formulation (p ≤ 0.05). Based on your suggested statistical analysis, figures 8, 9 and 12 have been replaced. On skin elasticity and on the wrinkle depth index, a better result seems to remain with the application of cream B. According to the new darta the text has been sligtly revised.

Point partially solved.

Please kindly provide the table results of the statistical evaluation in the supplementary material for better understanding of the results. Also to check if there is differences in other factors, such as volunteers, interactions between factors, etc. Line 796 “At T30, the hydration increase was 68.30% for A and 89.58% for B”. These values cannot be observed in figure 8 (I don’t know where the error is, in the plot or in the descriptive text). I encourage authors describe the result based on the mean (or median value, depending on the data distribution). And the discussion on lines 797 and 798 should be consider the mean values not the individual maximum o minim value.

Line 754-756. I disagree with author statement. First of all if they want to claim differences between formulation, a proper statistical analysis should be carried out, as I commented previously. According to box plot in figure 8, the mean value of T30 for creams A and B are similar, the discussion should be corrected. Authors should not magnify their results only for the maximum value, all data and the statistical analysis must be considered. According to the plot I have doubts about the superiority of cream B over A, and this could be explained by my previous comment about the composition of creams A and B. The same comments apply for al clinical studies carried out. A proper statistical method must be use and discussion must be based on the statistical results.

As you suggested a proper statistical analysis has been carried out. The discussion related to Figure 8 and other clinical results has been corrected, considering all data and the statistical analysis. According to the suggested statistical analysis the differences between A and B have been reconsidered.

Point previously discussed.

As requested,the authors send you in pdf format the tables with the raw results of the statistical evaluation of effect on skin parameters. The tables were exported directly from the program used for statistical processing (GraphPad Prism 7.00 for Windows).

Moreover, in the revised text the authors comment and discuss the results of the efficacy tests considering the mean values ​​registered at different measurement times for all volunteers, as they are now reported in the plots of Figures 8-12. The authors have removed the percentage of efficacy for each parameter. The authors agree that the percentage as previously calculated could confuse the reader. It was not the intention of the authors to magnify the effectiveness of B, but only to evaluate whether the new extract can contribute to the good performance of the new developed emulsion, without altering its stability over time.

We hope that the information now provided is clear and suitable to solve all your doubts
